# GENERALIZED BEHAVIOR LEARNING FROM DIVERSE DEMONSTRATIONS

**Varshith Sreeramdass, Rohan Paleja, Letian Chen, Sanne van Waveren, Matthew Gombolay**
Georgia Institute of Technology
{vsreeramdass, rpaleja3, letian.chen, sanne}@gatech.edu,
matthew.gombolay@cc.gatech.edu

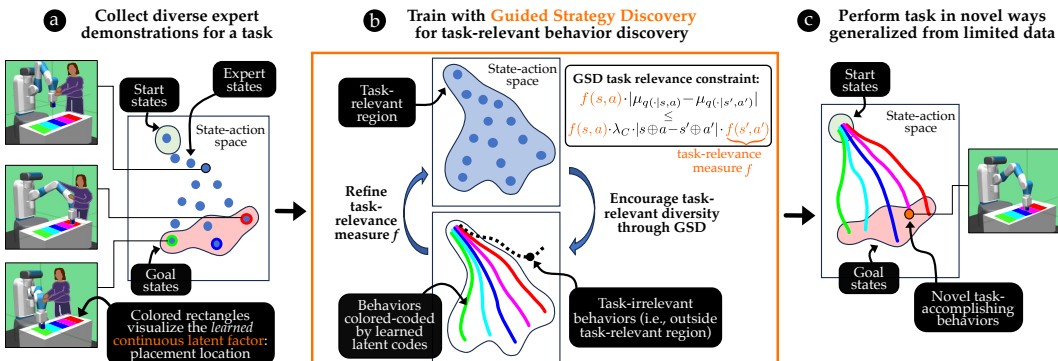

Figure 1: The figure overviews our framework: Guided Strategy Discovery. (a) Given diverse task demonstrations with underlying latent factors, (b) our framework optimizes a task-relevance guided diversity objective, (c) to discover behaviors that generalize to unseen latent factor values.

## ABSTRACT

Diverse behavior policies are valuable in domains requiring quick test-time adaptation or personalized human-robot interaction. Human demonstrations provide rich information regarding task objectives and factors that govern individual behavior variations, which can be used to characterize *useful* diversity and learn diverse performant policies. However, we show that prior work that builds naive representations of demonstration heterogeneity fails in generating successful novel behaviors that generalize over behavior factors. We propose Guided Strategy Discovery (GSD), which introduces a novel diversity formulation based on a learned task-relevance measure that prioritizes behaviors exploring modeled latent factors. We empirically validate across three continuous control benchmarks for generalizing to in-distribution (interpolation) and out-of-distribution (extrapolation) factors that GSD outperforms baselines in novel behavior discovery by $\sim$21%. Finally, we demonstrate that GSD can generalize striking behaviors for table tennis in a virtual testbed while leveraging human demonstrations collected in the real world. Code is available at github.com/CORE-Robotics-Lab/GSD.

## 1 INTRODUCTION

Intelligent agents that encounter a novel variation of a learned task should be able to adapt their default decision making to suit the variation at hand. To adapt on-the-fly, agents must learn a concise set of variations to quickly tune their behaviors. Such adaptability is valuable in applications such as few-shot learning (Duan et al., 2017) and personalized human-robot interaction (Wang et al., 2022) where limited examples or interaction must inform compatible approaches for task completion.

Behaviors that adaptable agents learn must be meaningfully diverse such that novel task variations can be addressed sufficiently. In the past, unsupervised reinforcement learning (RL) (Laskin et al., 2021) has been used to learn diverse behaviors or "skills". However, learned behaviors intended to explore the agent's environment may not directly be useful towards task completion. Such approaches are further limited by their inability to identify and exhibit meaningful variations that are useful during deployment. While supervised RL can be employed, reward specification to align diverse behaviors with user expectations can be cumbersome (Soares & Fallenstein, 2014).

In contrast to RL, learning from demonstration (LfD) methods enable agents to learn decision-making policies directly from human examples. The distinct variations[1] that individuals often exhibit, even when pursuing the same task objectives (Sanderson, 1989), reflect creative ways of task completion. We assume that these variations are governed by distinct but latent *behavior factors*. These factors impart useful diversity in human behaviors that adaptable agents can exploit.

Prior work in heterogeneous Imitation Learning (IL) (Li et al., 2017; Chi et al., 2023) largely focuses on generating behaviors corresponding to modes in training datasets or inferring representations of test behaviors. In this work, we address the challenge of generating behaviors with novel variations. We specifically study the ability to inter- and extrapolate from demonstrations to effectively produce new behaviors, that correspond to behavior factor values not seen in the training dataset, while also accomplishing the task. For example, consider a robot quadruped that runs at different speeds. We seek policies that run at 2m/s or 4m/s from demonstrations with speeds of 1m/s and 3m/s. Such a generalization ability can provide task-accomplishing behaviors with desirable characteristics directly through latent prior sampling. However, generalization with latent behavior factors is challenging, as we need to accurately identify the latent dimensions along which demonstrations vary and locate individual behaviors in the corresponding space before extending to novel behaviors.

We focus on novel behavior generation in the setting of online IL (Ho & Ermon, 2016) due to its data-sample efficiency. We show that prior approaches that utilize mutual information (MI)-based diversity objectives (Li et al., 2017) fail to produce novel behaviors. We draw inspiration from recent work in unsupervised RL (Park et al., 2022; 2024; 2023) that modify MI-based objectives and structure latent representations in order to induce specific behavioral traits (e.g., high Euclidean or temporal distances between states, controllability, etc). We propose to modify representation learning by restricting the latent space from capturing state-action space regions irrelevant to the task, identified through distillation of demonstration-specific occupancy measures. We find that our formulation encourages diversity specifically along traits that vary across demonstrations. We refer to this objective as task-relevant diversity as it produces behaviors that retain task-performance.

We present a novel approach to learn diverse task-accomplishing behaviors from demonstrations that generalize over latent behavior factors. Our contributions are four-fold:

- We show the need for a novel formulation of diversity for generalization in IL from diverse demonstrations through experiments in a 2D Point Maze domain (Sec. 4).
- We formulate task-relevant diversity, an objective to encourage diversity along factors of variation among demonstrations by restricting representations from capturing irrelevant regions. We propose Guided Strategy Discovery (GSD), an algorithm that optimizes diversity alongside imitation to discover novel task-accomplishing behaviors (Sec. 5).
- We demonstrate that GSD generalizes to novel behaviors with 21% average error reduction in behavior factors (known during evaluation) over four baselines across two splits (interpolation and extrapolation) in three domains spanning robot control, driving, and manipulation (Sec. 6.1).
- We demonstrate that GSD generalizes from physical human demonstrations to capture diverse stroke styles in a simulated Table Tennis domain (Sec. 6.3).

## 2 RELATED WORK

**Generalization in Behavior Learning** Prior works have studied generalization when agents are faced with task specifications from test distributions (Benjamins et al., 2023; Silva et al., 2021; Padalkar et al., 2023; Nair et al., 2022; Shridhar et al., 2023; Xu et al., 2022; Driess et al., 2020), or deployment settings different from training environments (Fu et al., 2018; Kumar et al., 2020; Packer et al., 2018; Kumar et al., 2021; Xie et al., 2023; Cobbe et al., 2019; Osa et al., 2022; Zahavy et al., 2022). In IL, generalization has been considered when demonstrators operate with diverse conditions (Qiu et al., 2023; Tangkaratt et al., 2020; Chen et al., 2021; Paleja et al., 2020; Schrum et al., 2023b; Li et al., 2017; Chen et al., 2020; Wang et al., 2017; Li et al., 2017; Hausman et al., 2017; Peng et al., 2022a). Our work focuses on the latter, where we study heterogeneous demonstrators with latent behavior factors. Among these, prior works either simply imitate multiple behaviors (Wang et al., 2017; Li et al., 2017; Hausman et al., 2017; Peng et al., 2022a), characterize heterogeneity through latent representations (Paleja et al., 2020; Schrum et al., 2023b; Li et al., 2017; Chen et al., 2020) or learn performant behaviors from diverse demonstrators (Qiu et al., 2023;

---

[1]Demonstration diversity can also be attributed to varying sub-optimal ways of performing a task (Ramachandran & Amir, 2007). Our work, however, focuses solely on task-optimal demonstrations.

Tangkaratt et al., 2020; Chen et al., 2021). Our work is the first to address learning both behavior representations and performant behaviors that generalize over behavior factors.

**Learning from demonstrations** Prior works in LfD utilize demonstrations to learn rewards (Abbeel & Ng, 2004; Ziebart et al., 2008; Fu et al., 2018; Chen et al., 2020; Ross et al., 2011) or task-performant policies (Ross et al., 2011; Ho & Ermon, 2016; Qiu et al., 2023; Tangkaratt et al., 2020; Chen et al., 2021). Our work seeks diverse policies, particularly from expert demonstrations with continuous latent factors (Wang et al., 2017; Li et al., 2017; Hausman et al., 2017; Chen et al., 2020; Peng et al., 2022a). While several works address heterogeneous IL with large datasets (Chi et al., 2023), we use environment interaction to tackle covariate shift in low data regimes (Ho & Ermon, 2016; Kostrikov et al., 2019; Reddy et al., 2020; Garg et al., 2021). Our work belongs to the class of adversarial IL (AIL) methods (Orsini et al., 2021) which model imitation as adversarial game between policies and a discriminator that captures expert occupancy. We study and address the limitations of MI-based diversity objectives used alongside AIL methods (Li et al., 2017; Hausman et al., 2017; Peng et al., 2022a) in capturing latent factors specific to demonstrations.

**Diverse behavior learning** Diverse behavior learning has been employed for exploration, pre-training, and generalization to novel environments. Quality Diversity (Batra et al., 2024) assumes the availability of task performance metrics and functions for measuring behavior factors. In contrast, our work is related to unsupervised RL (Laskin et al., 2021) that learn behaviors without such privileged information. Among them, we are similar to competence-based methods (Sharma et al., 2020; Hansen et al., 2019; Park et al., 2022; 2023; 2024) that learn latent spaces to represent heterogeneous behaviors. Works focus on different aspects of diversity, such as state coverage (Eysenbach et al., 2018; Park et al., 2022; Laskin et al., 2022; Mendonca et al., 2021), dynamics (Sharma et al., 2020), controllability (Park et al., 2023; 2024), etc. Our work adopts ideas of regularization (Park et al., 2022; 2023; 2024) for designing diversity objectives to improve heterogeneous IL.

**Structured methods for heterogeneous IL** CASSI (Li et al., 2023) uses MI-based objectives to learn novel locomotion behaviors from unlabeled data but relies on additional rewards for task completion, unlike our approach which does not depend on rewards. FLD (Li et al., 2024) structures latent spaces using differentiable fast Fourier transforms for periodic motions. In contrast, our approach is domain-agnostic, making it applicable across a broader range of tasks. ASE (Peng et al., 2022b) applies latent sequence modeling combined with hyperspherical priors for smooth motion transitions, and CALM (Tessler et al., 2023) builds on ASE with latent-conditioned discriminators. However, both methods suffer from limitations of naive MI formulations, which our work addresses.

## 3 PROBLEM STATEMENT AND PRELIMINARIES

We consider an infinite horizon, discounted, and reward-free Markov Decision Process (MDP\R), $(S, A, P, \rho_0, \gamma)$, where $S$ and $A$ represent state and action spaces, $P \colon S \times A \times S \to \mathbb{R}$, the transition probabilities, $\rho_0 \colon S \to \mathbb{R}$, the initial state distribution, and $\gamma$, the discount factor. An optimal expert policy $\pi^\xi$ is governed by a continuous factor, $\omega \in \Omega$. The factor space, $\Omega$, is split into **disjoint** train and test regions, $\mathrm{Tr}(\Omega)$ and $\mathrm{Te}(\Omega)$, respectively. Given a demonstrations set, $\mathscr{D}$, of trajectories $\tau_i^\xi = \{s_0, a_0, s_1, a_1, ...\}$, $a_t \sim \pi^\xi(\cdot|s_t, \omega_i)$, $s_{t+1} \sim P(\cdot|s_t, a_t)$, and $\omega_i \in \mathrm{Tr}(\Omega)$, we aim to learn a policy $\pi$ that captures the expert behavior $\pi^\xi$ over the entire factor space without access to $\omega_i$ or $\Omega$.

We ground our approach in InfoGAIL (Li et al., 2017), built upon Generative Adversarial Imitation Learning (GAIL) (Ho & Ermon, 2016) to imitate demonstrations: $J^{\mathrm{GAIL}} := E_{\pi^\xi}[\log D(s, a)] + E_\pi[\log(1 - D(s, a))]$, where $D$ is a discriminator that distinguishes between the learned policy, $\pi$, and the expert policy, $\pi^\xi$. InfoGAIL additionally introduces a latent variable, $z \in Z$, to capture factors underlying expert demonstrations. InfoGAIL optimizes MI by a variational lower bound (Barber & Agakov, 2004), $q(z|s, a)$. We refer to $q$ as the *decoder* as it infers $z$ from the state action pair. The InfoGAIL objective is $J^{\mathrm{InfoGAIL}} := J^{\mathrm{GAIL}} + \lambda_\mathrm{I} E_{z, \pi}[\log q(z|s, a)]$, where $\lambda_\mathrm{I}$ controls the diversity objective weight.

For formulating our diversity objective, we build on ideas from network distillation (Teh et al., 2017; Czarnecki et al., 2019; Chen et al., 2020). MSRD (Chen et al., 2020) learns task rewards from demonstrations, $\{\zeta_i\}$, over a finite set of factors by employing distillation with AIRL (Fu et al., 2018), a variant of GAIL that recovers a reward function, $r(s, a)$. MSRD then distills the reward functions for each variation, $r_{\zeta_i}$, into a common reward function for the task, $\tilde{r}_0$. The distillation is done by formulating each reward as, $r_{\zeta_i}(s, a) := \tilde{r}_0(s, a) + \tilde{r}_{\zeta_i}(s, a)$. The factor-specific residual reward, $\tilde{r}_{\zeta_i}(s, a)$, is encouraged to be close to zero with the additional objective,

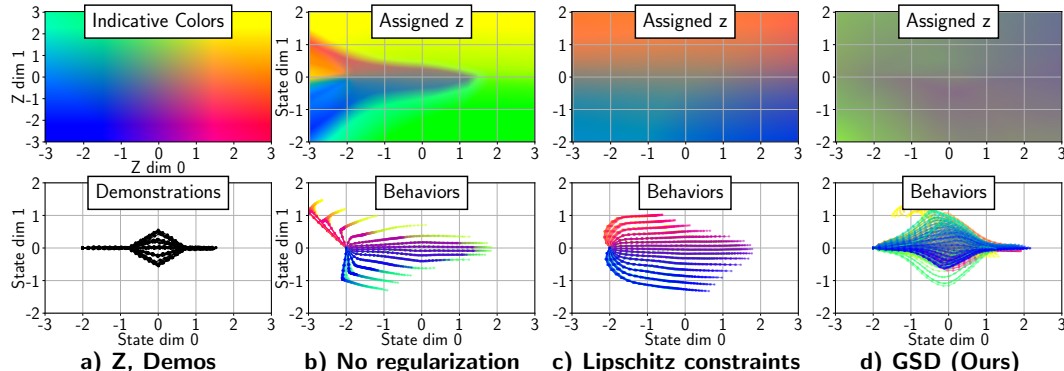

Figure 2: Fig. 2a (Top): Map shows colors assigned to the 2D latent space. Fig. 2a (Bottom): The agent starts at (-2, 0), moves to (2, 0), passing through (-1, 0), (0, $\omega$) and (1, 0) where $\omega \in [-1, 1]$. Full details are in Appendix A. Figs. 2b, 2c, 2d: Latent vectors assigned to the state space, with a state-only decoder, and policy behaviors, under a high importance weight $\lambda_I$, are visualized. Trajectories are colored according to conditioning latent vectors per Fig. 2a (Top). Baselines deviate from demonstrations arbitrarily (Fig. 2b, Bottom) or uniformly (Fig. 2c, Bottom), disregarding the goal. GSD (ours, Fig. 2d) discovers behaviors with novel latent variations (waypoints along $x = 0$) while reaching closer to the goal (2, 0).

$J^{\mathrm{MSRD}} := -E_{\zeta_i, \pi}[(\tilde{r}_{\zeta_i}(s, a))^2]$, to encourage the reward information common across demonstrations pertaining to the task to be represented by the task reward function $\tilde{r}_0$.

## 4 NEED FOR REGULARIZATION

In this section, we show that prior diversity formulations from Li et al. (2017); Park et al. (2022) fail to produce novel, task-accomplishing behaviors, motivating the need for a new formulation.

**No regularization:** InfoGAIL's diversity objective, MI, promotes diverse behaviors by rewarding the visitation of states associated with distinct latent vectors. In a 2D PointMaze domain with continuous state-action space (see Fig. 2), we show that an increased diversity objective's weight, $\lambda_I$, does not necessarily result in more diverse behaviors that accomplish the task. This result can be attributed to the decoder, $q$, a neural network (NN) that assigns latent vectors to state-action pairs, $\langle s, a \rangle$. Without regularization, the decoder, $q$, produces unconstrained latent assignments, with a high variety in smaller regions (see Fig. 2b, top, several distinct colors close to point (-2, 0)). This finding aligns with prior work (Choi et al., 2021; Park et al., 2022). Without regularization, related behaviors with close-by states can be mapped to unrelated far-away regions in the latent space without any meaningful structure. This behavior can cause insufficient (see Fig. 2b, bottom, several trajectories clump together along $y = 0$) or arbitrary (no pattern that governs deviation from demonstrations) behavior diversity.

**Prior regularization methods produce misaligned behaviors:** Prior works in unsupervised RL (Choi et al., 2021; Park et al., 2022) imposed Lipschitz constraints on the decoder, to enforce that for any two state-action pairs, $\langle s, a \rangle$, $\langle s', a' \rangle$, the assigned latent vectors (specifically the mean $\mu$ of the approximate posterior distribution $q(\cdot|s, a)$), differ by at most the Euclidean distance between the pairs, scaled by $\lambda_C$. Formally, $||\mu_{q(\cdot|s,a)} - \mu_{q(\cdot|s',a')}|| \leq \lambda_C \cdot ||s \oplus a - s' \oplus a'||$, where $\oplus$ denotes vector concatenation. The Lipschitz constraints ensure smooth latent vector assignments (see Fig.2c, top), which encourages behaviors to deviate from demonstrations uniformly over the state space. However, resulting behaviors do not necessarily proceed towards the goal (see Fig. 2c, bottom). Other diversity formulations focusing on controllability and temporal reachability (Park et al., 2023; 2024) will face similar issues if the auxiliary objectives are misaligned with behavior heterogeneity. We propose a general formulation that encourages behavior diversity along latent dimensions inferred from the demonstrations, without compromising task performance.

## 5 OUR METHOD: GUIDED STRATEGY DISCOVERY

We present our approach for achieving generalizable IL from diverse demonstrations.

## 5.1 Encouraging $f$-Relevant Diversity

First, we present a general approach for encouraging diversity selectively within state-action space regions indicated by high energy with respect to a scalar energy function, $f \colon S \times A \to [0, 1]$.

We design our approach by analysing transitions that occur during learning. Consider a scenario visualized in Fig. 3a, where an exploring agent is at a high $f$-energy state-action pair, $\langle s, a \rangle$, assigned a latent vector $z$, and it enters another pair, $\langle s', a' \rangle$. If a different vector, $z'$ s.t. $z' \not\approx z$, were assigned to $\langle s', a' \rangle$, the diversity rewards, $\log q(z|s, a)$, $\log q(z'|s', a')$, would encourage behaviors $\pi(\cdot|\cdot, z)$, $\pi(\cdot|\cdot, z')$, to visit $\langle s, a \rangle$ and $\langle s', a' \rangle$ respectively. The behavior, $\pi(\cdot|\cdot, z')$, would be desirable if $\langle s', a' \rangle$ has high $f$-energy, as it would visit a high energy pair different from $\langle s, a \rangle$, increasing coverage of high energy regions. On the other hand, if $\langle s', a' \rangle$ were a low $f$-energy pair, the behavior $\pi(\cdot|\cdot, z')$ visiting a low energy pair would be undesirable. In this case, the assignment for $\langle s', a' \rangle$ could be constrained close to $z$, which would remove the incentive for a behavior distinct from $\pi(\cdot|\cdot, z)$ to specifically visit $\langle s', a' \rangle$.

Selectively allowing distinct latent vector assignments only in high-energy regions encourages behaviors that target these regions, thereby promoting diversity only in high-energy regions. Constraint shown in Eq. 1 formalizes this intuition: For a transition from $\langle s, a \rangle$ to $\langle s', a' \rangle$, the latter's latent vector can be far from the former's by at most the Euclidean distance between the two pairs, scaled by $f$-energy of the latter and a factor $\lambda_C$.

$$||\mu_{q(\cdot|s,a)} - \mu_{q(\cdot|s',a')}|| \leq \lambda_C \cdot ||s \oplus a - s' \oplus a'|| \cdot f(s', a') \tag{1}$$

The proposed constraint (Eq. 1) disregards the energy of the starting state-action pair, $f(s, a)$. The constraint enforces the same latent assignment for pairs in a low→low energy transition and allows different latent assignments for pairs in a low→high energy transition as visualized in Fig. 3b. The enforcement can lead to connected low energy regions being assigned the same latent vector which is different from that of reachable high energy regions. Distinct latent vectors for low energy regions can result in behaviors visiting those low-energy regions, which is **not** desirable.

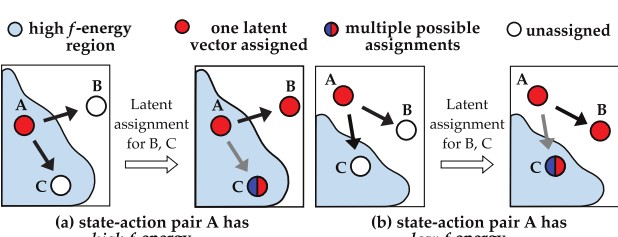

(a) state-action pair A has *high f*-energy     (b) state-action pair A has *low f*-energy

Figure 3: This figure shows a visualization of state transitions that an agent at state-action pair, A, may undergo and the effect of our constraint in Eq. 1. The light blue region indicates high $f$-energy and white state-action pairs do not yet have assigned latent vectors. The lightness of the arrows indicates the slackness of our constraint (Eq. 1) from scaling with $f$-energy on the right-hand side.

We rectify our constraints to prevent latent assignments for low energy regions, by enforcing them only with transitions with high energy starting pairs through scaling the constraint with $f$-energy of the starting pair, as shown in Eq. 2. Thus, when the starting pair is of low energy, the constraint implemented with a Lagrange multiplier is less effective due to a smaller violation.

The modified constraints encourage the decoder to effectively use the latent space to solely represent high-energy regions. We refer to the resulting objective as $f$-relevant diversity.

$$f(s, a) \cdot \left[ ||\mu_{q(\cdot|s,a)} - \mu_{q(\cdot|s',a')}|| \right] \leq f(s, a) \cdot \left[ \lambda_C \cdot ||s \oplus a - s' \oplus a'|| \cdot f(s', a') \right] \tag{2}$$

## 5.2 Inferring a Task-Relevance Measure from Demonstrations

Diverse task-accomplishing behaviors can be learned if an appropriate energy function $f$ can be used to instantiate our $f$-relevant diversity objective. We now present an approach to derive such an $f$ from demonstrations, utilizing occupancies captured by the discriminator, $D$.

Function $f$ should indicate regions where agent occupation is favorable for the task while also capturing demonstrators' heterogeneity. We classify these regions as: (I) regions occupied by all experts, and (II) subspaces where different experts occupy distinct smaller regions. We propose that capturing these two types of regions in the energy function provides us with an objective that encourages task accomplishment and generalization over behavior factors.

While the discriminator, $D$, could be used to model $f$, it would capture type II subspace insufficiently. $D$ is trained to capture the union of demonstration occupancies, covering all type I regions,

---

**Algorithm 1** Guided Strategy Discovery

---

**Input**: Dataset of diverse expert demonstrations, $\mathscr{D} = \{\tau_i^\xi\}$
**Output**: Latent-conditioned policy capturing diverse behaviors, $\pi$

1: Initialize policy $\pi$, task relevance $f$, factor-specific residual $g$, bias $b$, and decoder $q$
2: **for** $i \in \{0, 1, 2, ...\}$ epoch **do**
3:  Sample $z^\pi$ from prior, $\tau^\pi$ using policy $\pi(\cdot|\cdot, z^\pi)$; $\tau^\xi$ from $\mathscr{D}$, infer $z^\xi$ using decoder $q$
4:  Compute discriminator outputs according to the conditioned distillation structure (Eq. 3):
   $D(s, a, z) = \sigma(\lambda_S \cdot [f(s, a) + g(s, a, z)] + b)$
5:  Update $f$, $g$, $b$ with gradient ascent on the discriminator objective computed by linearly combining $J^{\text{GAIL}}$ (Ho & Ermon, 2016) and the distillation objective (Eq. 4):
   $\mathbb{E}_{\tau^\xi}[\log D(s, a, z^\xi)] + \mathbb{E}_{\tau^\pi}[\log(1 - D(s, a, z^\pi))] - \mathbb{E}_{\tau^\xi}[(g(s, a, z^\xi))^2] - \mathbb{E}_{\tau^\pi}[(g(s, a, z^\pi))^2]$
6:  Compute the decoder objective using policy behavior samples (Eysenbach et al., 2018):
   $\mathbb{E}_{\tau^\pi}[\log \mathcal{N}(z^\pi|\mu_{q(\cdot|s,a)}, \Sigma_{q(\cdot|s,a)})]$
7:  Update decoder with gradient ascent on the objective while enforcing task-relevance constraints (Eq. 2): $[\lambda_C \cdot ||s \oplus a - s' \oplus a'|| \cdot f(s', a') - ||\mu_{q(\cdot|s,a)} - \mu_{q(\cdot|s',a')}||] \cdot f(s, a) \le 0$
8:  Update policy $\pi$ with RL using linearly combined behavior imitation and diversity rewards, $\log(D(s, a, z^\pi))$ and $\log \mathcal{N}(z^\pi|\mu_{q(\cdot|s,a)}, \Sigma_{q(\cdot|s,a)})$, respectively.
9: **end for**

---

but only certain regions in the type II subspace that correspond to training demonstrations. $f$ modeled in this way would limit behavior discovery beyond the training dataset.

We employ the distillation of demonstration-specific occupancy measures into a common measure, to model $f$ and capture type II subspaces. Demonstration-specific occupancies are obtained by conditioning the discriminator, $D(s, a, z)$, on the inferred latent vector, $z$. It is further split as a linear combination of a latent-independent measure, i.e., our desired energy function, $f(s, a)$, and a dependent term, $g(s, a, z)$, in the logit space as shown in Eq. 3, where $g\colon S \times A \times Z \to [0, 1]$, $\sigma$ is the logistic function, $\lambda_S$, a scaling constant, and $b$, a learnable bias. $\lambda_S$ and $b$ are introduced to transform the sum of bounded measures and enable $D$ to capture most of the probability range $[0, 1]$. The discriminator is trained with an additional distillation objective, as shown in Eq. 4, to minimize the residual, $g$, to only capture demonstration-specific occupancy.

$$D(s, a, z) = \sigma(\lambda_S \cdot [f(s, a) + g(s, a, z)] + b) \tag{3}$$

$$J^{\text{R}} := -E_{z,\pi}[(g(s, a, z))^2] \tag{4}$$

The objective, $J^{\text{R}}$, encourages $f$ to fully capture both type I and II regions. Type I regions are captured by $f$, as $g$ is driven to zero where occupancy is common across demonstrations and latent-independent. $g$ is encouraged to be close to zero even in regions with demonstration-specific occupancy, causing it to capture minimal possible information and distilling the rest into $f$. We posit that $f$ indicates entire subspaces where occupancy is demonstration dependent, i.e., type II subspaces, while $g$ indicates regions in these subspaces specific to each demonstration. We call our procedure for deriving $f$ conditioned distillation (ConDist), due to its use of latent conditioning and distillation, similar to prior reward distillation frameworks (Chen et al., 2020).

**Algorithm:** We combine $f$-relevant diversity and conditioned distillation to propose Guided Strategy Discovery (GSD). High-level steps are outlined in Algorithm 1. Complete details are in Appendix C. In each epoch, we sample behaviors using the policy conditioned on latent vectors from the prior (Line 3). Latent vectors for demonstrations are inferred using the decoder (Line 3). We use the proposed discriminator structure (Line 4), define the imitation and distillation objectives and update the energy function, residual, and bias, using gradients (Line 5). We optimize the variational lower bound (Line 6) while enforcing with proposed constraints (Line 7) to update the decoder using gradients. Finally, we update the policy with rewards from the discriminator and decoder (Line 8).

We posit that $f$-relevant diversity and conditioned distillation are synergistic. An accurate $f$ function representing demonstrations can guide latent assignments and associated behaviors to generalize beyond demonstrations. A latent space representing diverse demonstrations can help distillation capture regions beyond demonstrations that generalize latent behavior factors. Fig. 2d (bottom) shows that with GSD, the learned behaviors in 2D PointMaze capture novel latent variations by passing through waypoints along $x = 0$ while reaching closer to the goal of (2, 0) better than baselines in Figs. 2b, 2c. In addition, GSD produces a higher fraction of goal-reaching trajectories despite a low weight for the imitation objective and a weaker incentive to match the expert.

## 6 EVALUATION

We present empirical evaluation to answer the following research questions:

1. How do various methods perform in generalization to behaviors with novel latent factors while maintaining task performance? (Sec. 6.1)
2. How do various methods structure behaviors in the latent space? (Sec. 6.2)
3. How do various methods perform in learning diverse task-accomplishing behaviors from real-world human demonstrations? (Sec. 6.3)

**Domains:** Existing benchmark, D3IL (Jia et al., 2024), focuses on discrete behavior modes. We instead use domains with continuous variation and clear task objectives to evaluate generalization from limited demonstrations. For Sec. 6.1, 6.2, we use HalfCheetah (Wawrzyński, 2009), Fetch-PickPlace (Plappert et al., 2018) and DriveLaneshift (Leurent, 2018) as they provide well-defined tasks with distinct one-dimensional (1D) factors. We script expert policies based on these factors. In HalfCheetah, the robot runs at various speeds; in FetchPickPlace, the arm places an objects at different locations; and in DriveLaneshift, the ego-car overtakes at varying headway distances. 1D factors help avoid multiple sources of heterogeneity allowing careful examination of learned policies.

**Methods:** We consider InfoGAIL as our base method, representative of approaches that combine online IL with MI-based diversity. Heterogeneous online IL methods capture finite sets of behaviors (Chen et al., 2020) are omitted due to less scope for novel behavior discovery. While we incorporate some improvements from adversarial IL (Orsini et al., 2021) across all the evaluated methods, a thorough evaluation of their integration with diversity objectives is left for future work. Comparison with other online IL methods (Garg et al., 2021) is omitted as they do not directly address demonstration diversity. Comparison with offline IL approaches that learn without environment interaction is presented in Appendix D. We compare the following variants of InfoGAIL.

- IG: InfoGAIL (Li et al., 2017) with a continuous two dimensional latent variable.
- IG+Lipz: IG with Lipschitz constraints for decoder $q$ to investigate the uniform diversity.
- IG+Con: IG with a conditioned discriminator $D(s, a, z)$ structure to investigate the effect of conditioning the discriminator.
- IG+ConDist: IG with our proposed conditioned distillation to investigate the effect of extraction of a task-relevance measure (Eqs. 3, 4).
- IG+ConDist+Lipz: IG+ConDist with Lipschitz to investigate the uniform diversity formulation alongside conditioned distillation.
- **GSD (Ours)**: IG+ConDist with our proposed task-relevant diversity formulation (Eq. 2).

### 6.1 QUANTITATIVE EVALUATION

We investigate whether learned behaviors can represent factor values in the disjoint test region $\text{Te}(\Omega)$, after training on demonstrations, $\mathscr{D}$, from the train region, $\text{Tr}(\Omega)$ (see Sec. 3). We consider factors that are measurable from trajectories (only for the sake of evaluation) to assess recovery performance, i.e., how well the learned latent space can represent expert behavior, by comparing desired and measured factor values. When diverse expert behaviors form distinct modes, this framework checks if continuous factors underlying them can be accurately identified and generalized.

**Splits:** We divide the bounded 1D factor range into five consecutive equal-sized intervals:

- **Interpolation**: The first, third, and fifth intervals represent the train region, and the second and fourth are the test region. The split allows us to evaluate the ability to interpolate behaviors to two factor space intervals while providing three non-consecutive intervals to represent the factor.
- **Extrapolation**: The second and fourth intervals represent the train region, while the first and fifth intervals are the test region. We choose two non-consecutive intervals for the train region to have a sparse dataset while providing enough diversity to represent the factor.

These splits evaluate how well the latent space captures factors to interpolate and extrapolate behaviors. We use five demonstrations per interval (details in Appendix B).

**Metrics:** We search for desired behaviors using $K \in \{10, 20, 30, 40, 50\}$ latent vector samples from the prior $p_z(\cdot)$, where $K$ represents the test time search sample-complexity, varied to investigate how well we generate desired behaviors from limited samples. We roll out policies conditioned on the sampled vectors, measure the factors of the sampled behaviors, and compute the least mean absolute error (MAE) between the desired and the $K$ measured values, averaging over $1500/K$ rounds. We

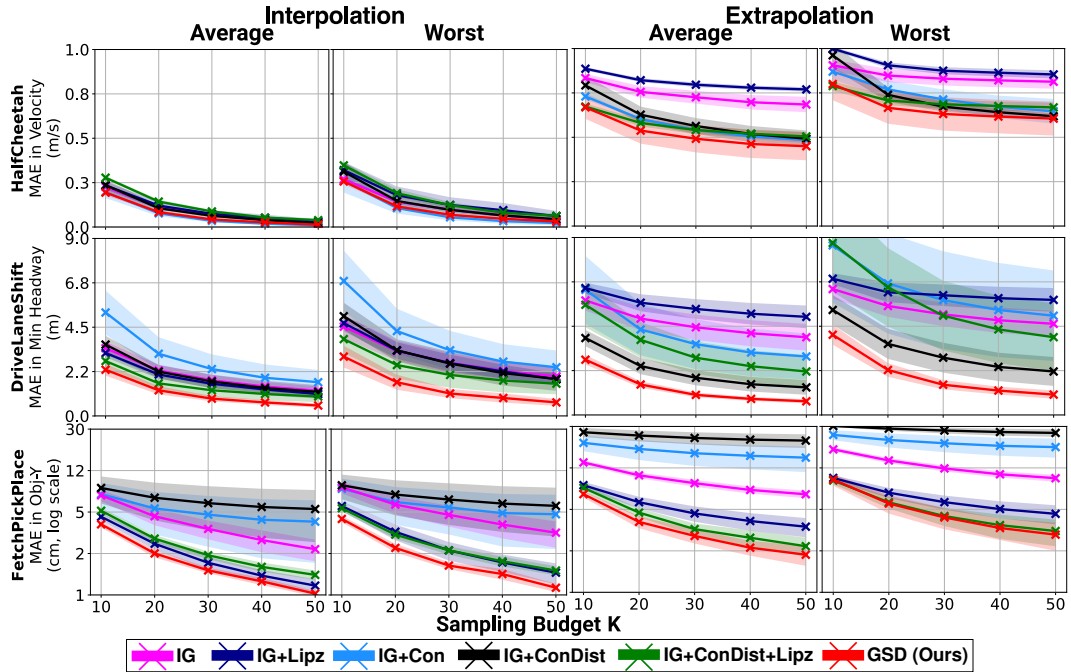

Figure 4: The figure shows average and worst-case recovery errors over the three domains and two factor space splits corresponding to interpolative and extrapolative generalization. Shaded regions are standard errors over five train seeds. GSD outperforms baselines in recovery of unseen latent factors across most domains and splits.

refer to this metric as the **recovery error**. We consider the midpoints of test intervals as the set of desired values. We report average and worst errors over desired values in the test region, providing estimates of closeness between desired values and closest available behavior's factor, on average and worst case. We report mean and standard errors over five train seeds in Fig. 4. We evaluate task performance by averaging environment returns over 1500 latent samples. We show recovery and task performance tradeoff in Fig. 5. Exact numbers are provided in Appendix H.

**Lipschitz constraints:** For HalfCheetah (Fig. 4, top row), IG+Lipz and IG+ConDist+Lipz have worse recovery errors compared to IG and IG+ConDist, indicated by the dark blue curve above magenta, and green above black, respectively. For DriveLaneshift (middle row), IG+Lipz and IG+ConDist+Lipz exhibit the same trend against IG and IG+ConDist: for interpolation, errors are improved shown by dark blue falling below magenta and green below black; and for extrapolation, the errors are worsened. For FetchPickPlace (bottom row), IG+Lipz and IG+ConDist+Lipz improve over IG and IG+ConDist indicated by dark blue and green consistently below magenta and black respectively. Lipschitz constraints seem to be benefiting FetchPickPlace alone, which might be due to "uniform diversity" aligning with object-

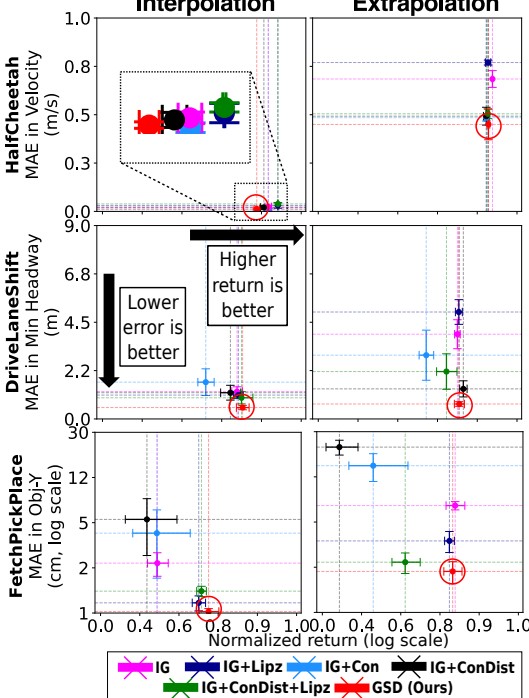

Figure 5: The figure shows the tradeoff between task and recovery performance for three domains and two splits. Error bars show standard errors over five seeds. High returns (x-axis) and low errors (y-axis) are better. GSD (circled in red) improves recovery while retaining or improving task performance across most domains and splits.

position factors, that is absent in other domains. This supports our hypothesis that relevant factors for diversity must be inferred from demonstrations to benefit all domains.

**Conditioning:** For HalfCheetah (top row), IG+Con improves errors compared to IG, indicated by the light blue curve below magenta. However, DriveLaneshift and FetchPickPlace, IG+Con seems to worsen performance, with light blue largely above magenta in the bottom two rows. Worsened errors may be a result of conditioning on a latent variable capturing arbitrary factors.

**Conditioned Distillation:** For HalfCheetah (top row), IG+ConDist and IG+ConDist+Lipz improve over IG and IG+Lipz for extrapolation, indicated by black and green curves below magenta and dark blue respectively. They remain on par for interpolation. For DriveLaneshift as well (middle row), IG+ConDist and IG+ConDist+Lipz improve over IG and IG+Lipz for extrapolation ($K \geq 30$) and remain on par for/slightly improve interpolation. For FetchPickPlace (bottom row), the trends are interesting. IG+ConDist worsens errors over IG for interpolation and extrapolation, indicated by black above magenta. However, with Lipz's addition, IG+ConDist+Lipz tends close to IG+Lipz for interpolation and outperforms it for extrapolation. The patterns firstly suggest that conditioned distillation can improve extrapolation performance. In addition, for FetchPickPlace where Lipschitz constraints are particularly effective, conditioned distillation can further improve extrapolation.

**Task-relevant Diversity:** GSD improves recovery over other approaches across most domains and splits, shown by the red curve below others in all plots, except for interpolation with HalfCheetah (top row, first two columns). In HalfCheetah (top row), the close performance across methods may be attributed to wide differences in gait styles across velocities that are challenging to interpolate or extrapolate. In DriveLaneshift (middle row), GSD reduces recovery error considerably over other approaches. In FetchPickPlace, GSD is closely matched by IG+Lipz or IG+ConDist+Lipz as Lipschitz constraints already capture relevant factors. Nevertheless, GSD can further improve recovery.

**Tradeoff between Task and Recovery Performance:** Across all domains, GSD either matches or improves average normalized returns over the latent prior, as indicated in Fig. 5 by the red cross generally being aligned with or positioned further right than others in all domain-split combinations but one. These results demonstrate the effectiveness of GSD's task-relevant diversity formulation in learning behaviors that reduce recovery error while maintaining task performance.

## 6.2 QUALITATIVE EVALUATION

We visualize the nature of the learned latent spaces for extrapolation in FetchPickPlace in Fig. 6. IG+ConDist learns a large behavior set for placing the object in the red test region (indicated by red cells in Fig. 6b), but ignores the dark-blue test region. While IG+ConDist+Lipz learns behaviors for all regions, it learns several that fail to place the object quickly enough (indicated by white cells in Fig. 6c). GSD learns behaviors that achieve the task (few white cells in Fig. 6d) while representing all place locations equally in proximity to each other (roughly equal number of cells across colors nearby each other). GSD exhibits potential for improving the accountability of policy learning by enabling well structured latent spaces.

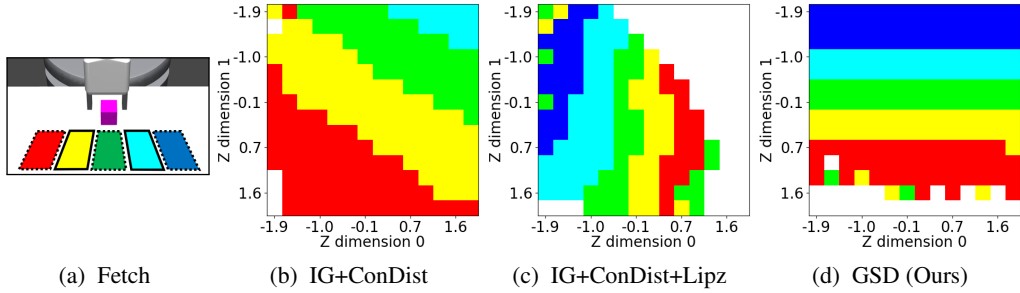

|  (a) Fetch | (b) IG+ConDist | (c) IG+ConDist+Lipz | (d) GSD (Ours) |

Figure 6: Fig.6a shows FetchPickPlace with object placement locations color-coded. Solid and dotted boundaries indicate train and test regions respectively. Figs.6b,6c,6d: Policy behaviors are shown in the 2D latent space through colors for resulting place-locations shown in 6a. White regions indicate failed placements or placements with low task reward. Behaviors with IG+ConDist (6b), IG+ConDist+Lipz (6c) either represent the relevant regions disproportionately or fail to accomplish the task. **Behaviors with GSD (6d) accomplish the task (low presence of white cells) and represent all regions well (roughly equal number of cells across colors).**

## 6.3 EVALUATION WITH REAL-WORLD HUMAN DEMONSTRATIONS

We further evaluate our approach to test scalability to complex tasks with human demonstrations in a Table Tennis (TT) domain. TT represents a dynamic domain that requires precise robot motion and fast reaction times while acting on noisy observations. Our physical setup consists of a Barrett WAM Arm mounted to the ceiling in front of a TT table, and a racquet attached as the arm's effector. Balls are launched using a Butterfly Amicus launcher at a

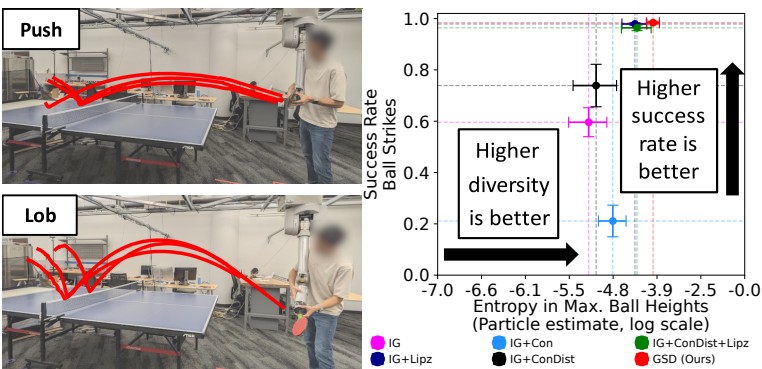

Figure 7: **Left:** The images visualize ball trajectories achieved by an expert kinesthetically demonstrating two types of strokes. **Right:** The figure shows the tradeoff between ball striking success rate and diversity in ball heights achieved. GSD outperforms baselines for both metrics.

fixed orientation and velocity with some noise. Balls are detected and tracked using a YOLO object detector and a Kalman Filter. An expert provides kinesthetic demonstrations of push and lob strokes. We recreate the setup in simulation with PyBullet for behavior learning. Ball initialization and observation noise levels in the simulation match real data. Complete details are in Appendix E.

While multiple continuous factors may exist underlying TT stroke styles, we evaluate generalization for maximum ball height, which we assume to be one of the underlying continuous factors. We evaluate various methods in simulation for achieving high diversity in ball heights. We compute entropy in ball height values using particle estimates (Singh et al., 2003), after disregarding unsuccessful trials that fail to strike the ball over the table. We report the success rate traded off with diversity in ball heights in Fig. 7 (right). Our method GSD outperforms all baselines in both measures of success rate and entropy.

## 7 CONCLUSION, LIMITATIONS AND FUTURE WORK

We study the problem of generalization from diverse demonstrations over underlying latent factors. We investigate the shortcomings of prior MI-based methods and propose a novel diversity formulation. Our empirical evaluation shows that our approach improves the recovery of factors over the next best baseline (for $K$=50) by 18.3% and 24.6% for interpolation and extrapolation respectively while retaining task performance in three domains with synthetic demonstrations. Our qualitative analysis shows our potential in improving the accountability of learned policies. Lastly, our experiments with real-world human demonstrations shows that our framework can capture a diverse range of task-accomplishing behaviors in a challenging domain requiring quick response times.

**Limitations:** Our experiments focus on demonstrations with one-dimensional latent factors. Our approach may struggle with higher dimensional or non-Markovian factors, which could require specialized designs for disentangling dimensions or capturing observation history dependence. Scaling to visual domains, where continuous factors must be inferred from sparsely distributed demonstrations, may also be challenging, as simple architectures for the energy function $f$ may not generalize well. Furthermore, our assumption that demonstration occupancies correlate with task success may be violated with non-expert demonstrators or partial observability, which may require state estimation models. While current work prioritizes validating our core contributions, we plan to evaluate scalability in future work.

Our evaluation with human demonstrations is further limited to quantitative metrics. We aim to conduct user studies to subjectively evaluate behaviors in human robot interaction settings. Our evaluation is further limited to simulated domains. We aim to explore the efficacy of our diversity formulation for learning novel behaviors in physical robot systems with improved data-sample efficiency. We further aim to explore the theoretical implications of our formulation and its alignment with the imitation objective. Further limitations are discussed in Appendix F.

## REPRODUCIBILITY STATEMENT

Data generation and collection is detailed in Appendices A, B, E.1. Implementation details, hyperparameters and evaluation procedures are detailed in Appendices C, E.2. Code is available at github.com/CORE-Robotics-Lab/GSD.

## ACKNOWLEDGMENTS

We thank Manisha Natarajan for feedback on the manuscripts. This work was supported by MIT Lincoln Laboratory grant FA8702-15-D-0001, NIH grant 1R01HL157457-01A1, NSF grant CPS-2219755, ONR grant N00014-22-1-2834, and a grant from Ford Motor Company.

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

## A  POINT MAZE

The PointMaze domain considered in Sec. 4 is presented in Fig. 8. PointMaze is a two-dimensional navigation environment with continuous state and action spaces. The state vector represents the agent's current location's x- and y- coordinates in $[-\infty, \infty]^2$. The action space is a velocity command, a two-dimensional vector in $[-1, 1]^2$. The episode length is fixed to 25 steps. Expert demonstrations are collected from a PD controller parameterized with a one-dimensional (1D) factor, $\omega$ that determines the waypoint through which the agent passes $(0, \omega)$ on its way to the goal $(2, 0)$.

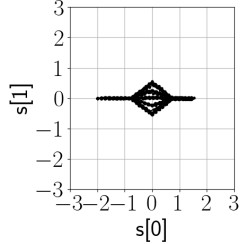

Figure 8: The figure visualizes expert demonstrations in PointMaze with varying waypoints along $x = 0$.

## B  DOMAINS AND DEMONSTRATIONS

The bounded factor range is divided into 5 intervals for each domain, as explained in Sec. 6. For each interval, we add Gaussian noise to the mean value of the interval to generate five samples. We condition the expert policy on the five samples to obtain five demonstrations for each interval.

### B.1  HALFCHEETAH

The HalfCheetah environment considered in Sec. 6 is from OpenAI Gym (Brockman et al., 2016). The observation vector consists of the positions and velocities of the robot joints, along with height and velocities in the vertical and horizontal directions. The reward is modified as shown in Eq. 5, where $r_t$ is the reward at the time, $t$,

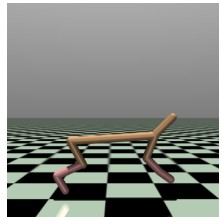 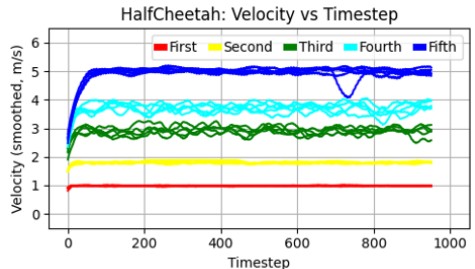

Figure 9: **Left:** The images visualizes the HalfCheetah robot. **Right:** The figure shows the (smoothed) velocity plotted against the timestep of the demonstration. The trajectories are colored to indicate the factor interval in which they belong.

and $x_t$ is the position of the center of mass of the robot along the x-axis at time $t$, and $I$, the indicator function that outputs 1 if and only if (iff) its argument evaluates to true. The undiscounted episode return counts the number of steps in which the cheetah moves forward by a non-zero amount. The return is normalized using the range, $[0, 1050]$. The episode length is fixed at 1000 steps. The environment is stochastic with the robot initialized at random configurations.

$$r_{step} = I(x_{t+1} - x_t > 0) \tag{5}$$

The factor is the mean velocity measured as the net change in the x-coordinate over the elapsed time. Due to environment stochasticity, we use five sampled trajectories per conditioning latent vector during evaluation and consider the mean value. Demonstrations consist of the robot running at different mean velocities $[1, 2, 3, 4, 5]$ m/s, collected using RL policies trained using SAC (Haarnoja et al., 2018) and auxiliary rewards for target velocities.

### B.2  DRIVELANESHIFT

The DriveLaneshift environment is built from the highway-env library (Leurent, 2018). The highway consists of two lanes. The scenario includes the ego-car in the right lane controlled by the agent, and another car in front, in the same lane, that maintains a constant speed of 25 m/s. The task of the ego-car is to shift to the left lane, overtake the other car, and reach the target speed of 30 m/s. The reward at each step is as shown in Eq. 6, where $r_t$ is the reward at the time, $t$, $b_{onroad}$, evaluates to true iff the car is within the road bounds at time $t$, $b_{safe}$ evaluates to true iff the ego-car has not crashed until time $t$, $b_{leftlane}$ evaluates to true iff the ego-car is in the left lane at time $t$, $v_t$ is the speed at time $t$, and $clip(x, a, b)$ clips the value $x$ to lie between $a$ and $b$. The return is normalized

using the range $[0, 175]$. The state vector includes positions (absolute for the ego-car, relative for the other), velocities, heading angles, and longitudinal, latitudinal, and angular offsets to the closest lane for both cars. The episode length is fixed at 50 steps. The environment is deterministic.

$$r_{step} = I(b_{onroad}) + I(b_{safe}) + I(b_{leftlane}) + clip(\frac{|v_t - 30|}{5}, 0, 1) \tag{6}$$

The factor is the min headway distance, i.e., the distance between the ego-car and the other, at which the ego-car shifts to the left lane before overtaking. Demonstrations consist of the ego-car performing overtaking maneuvers at varying min headway distances $[10.92, 18.28, 25.62, 32.91, 40.27]$ m, collected using a scripted PD controller.

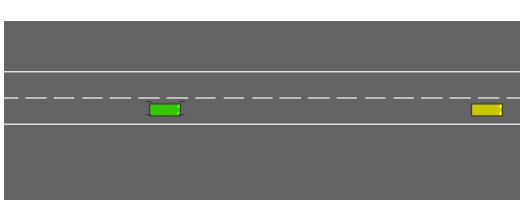
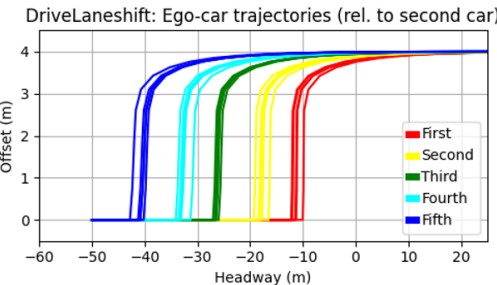

Figure 10: **Top:** The images visualize the highway overtaking scenario with the ego-car (green) starting behind the other car (yellow) in the right lane. **Bottom:** The figure visualizes the position of the ego-car relative to the other car as recorded in the demonstrations. The trajectories are colored to indicate the factor interval in which they belong.

## B.3 FETCHPICKPLACE

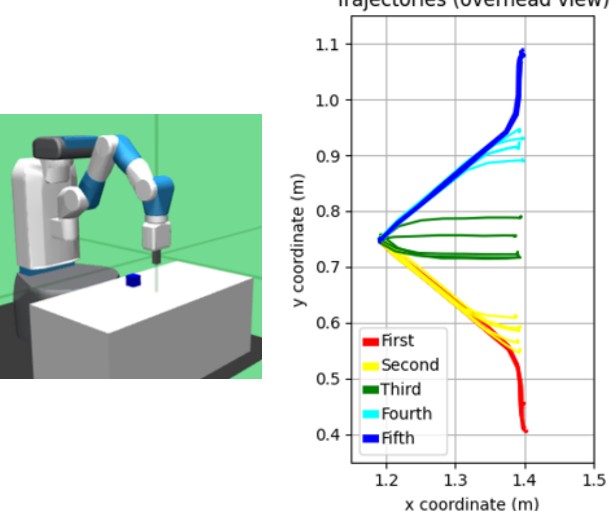

The FetchPickPlace environment considered is from the gym library (Brockman et al., 2016). The task is to move the object from its initial location on the table at $[1.20, 0.75]$ to along the line $x = 1.40$, with reward at each step measured as shown in Eq. 7, where $r_t$ is the reward at the time, $t$, and $x_t$ is the position of the center of mass of the object along the x-axis at time $t$. The return is normalized using the range $[-20, -5]$. The state vector includes the end effector position and velocity, object position and velocity, finger gripper position and velocity, and object position relative to the gripper. The episode length is fixed at 100 steps. The environment is deterministic.

$$r_{step} = -|x_{t+1} - x_t| \tag{7}$$

Figure 11: **Left:** The images visualize the Fetch robot with an object on the table. **Right:** The figure shows the object trajectories (from the top down) recorded in the demonstrations. The trajectories are colored to indicate the factor interval in which they belong.

The factor is the y-coordinate of the final object position. Demonstrations consist of the robot arm picking the object up from the initial location and placing it at the target x-coordinate and varying y-coordinates, $0.75 + [-0.32, -0.16, 0, 0.16, 0.32]$ m, collected using a scripted state-based PD controller.

# C  Algorithm and Implementation

## C.1  Algorithm - Detailed Version

A detailed version of Algorithm 1 can be found in Algorithm 2, with objectives and gradient steps for all components explicitly written down.

---

**Algorithm 2** Guided Strategy Discovery

---

**Input**: $\mathscr{D} = \{\tau_i^\xi\}$
**Output**: $\pi$

1: Initialize policy $\pi$, task relevance $f$, factor-specific residual $g$, decoder $q$, with parameters $\theta_\pi$, $\theta_f, \theta_g, \theta_q$, Lagrange multiplier $\lambda$, bias $b$, and learning rates $\eta_\pi, \eta_f, \eta_g, \eta_q, \eta_\lambda, \eta_b$
2: **for** $i \in \{0, 1, 2, ...\}$ epoch **do**
3:     Sample $z^\pi$ from prior, $\tau^\pi$ using policy $\pi(\cdot|\cdot, z^\pi)$; $\tau^\xi$ from $\mathscr{D}$, infer $z^\xi$ using decoder $q$
4:     Define objective for functions $f$, $g$ and bias $b$:
$$D(s, a, z) = \sigma(\lambda_S \cdot [f(s, a) + g(s, a, z)] + b)$$
$$J^{\mathrm{I}} \leftarrow \mathbb{E}_{\tau^\xi}[\log D(s, a, z^\xi)] + \mathbb{E}_{\tau^\pi}[\log(1 - D(s, a, z^\pi))]$$
$$- \mathbb{E}_{\tau^\xi}[(g(s, a, z^\xi))^2] - \mathbb{E}_{\tau^\pi}[(g(s, a, z^\pi))^2]$$
5:     Update $f$, $g$, $b$ using gradients: $[\theta_f, \theta_g, b] := [\theta_f, \theta_g, b] + [\eta_f \nabla_{\theta_f} J^{\mathrm{I}}, \eta_g \nabla_{\theta_g} J^{\mathrm{I}}, \eta_b \nabla_b J^{\mathrm{I}}]$
6:     Define objective for decoder $q$:
$$\delta(s, a, s', a') \leftarrow [\lambda_C \cdot ||s \oplus a - s' \oplus a'|| \cdot f(s', a') - ||\mu_{q(\cdot|s,a)} - \mu_{q(\cdot|s',a')}||] \cdot f(s, a)$$
$$q_L(z, s, a) \leftarrow \mathcal{N}(z|\mu_{q(\cdot|s,a)}, \Sigma_{q(\cdot|s,a)})$$
$$J^{\mathrm{E}} \leftarrow \mathbb{E}_{\tau^\pi}[\log q_L(z, s, a) + \lambda \cdot \min(\delta(s, a, s', a'), \epsilon)]$$
7:     Update decoder $q$ and $\lambda$ using gradients: $[\theta_q, \lambda] := [\theta_q, \lambda] + [\eta_q \nabla_{\theta_q} J^{\mathrm{E}}, -\eta_\lambda \nabla_\lambda J^{\mathrm{E}}]$
8:     Update policy $\pi$ with RL using rewards: $r(s, a, z) = \log(D(s, a, z)) + \lambda_{\mathrm{I}} \cdot \log q_L(z, s, a)$
9: **end for**

---

## C.2  Implementation

We implement our approach on top of the public code-base for VILD (Tangkaratt et al., 2020) that implements adversarial IL algorithms using PyTorch (Imambi et al., 2021): github.com/voot-t/vild_code. We use implementation tricks from their codebase to ensure convergence across methods, such as gradient penalty (Gulrajani et al., 2017) with a weight of 0.1 for the discriminator/task relevance, and the positive logarithmic function (Wang & Li, 2021) for discriminator rewards, i.e., $r(s, a) = -\log(1 - D(s, a))$ instead of $r(s, a) = \log(D(s, a))$.

We use a normal prior for the latent space across all approaches. The decoder $q$ outputs the mean and diagonal elements of the covariance matrix of the approximate posterior distribution, which is assumed to be Gaussian.

**Conditioned Discriminator** To infer latent code $\tau^\xi$ for a demonstration trajectory $\tau^\xi$, we make a simplifying assumption that the posterior distributions across transitions are independent. Thus, the product of the individual distributions gives us the posterior distribution for the demonstration trajectory.

We add expert transitions with mismatched demonstration latent vectors as fake samples to the discriminator dataset to ensure that the conditioned discriminator, $D(s, a, z)$, does not ignore the input latent vector. We upsample "real" data points in a batch to avoid imbalanced classes for discriminator gradient updates.

**Decoder Regularization** We perform spectral normalization (Miyato et al., 2018) using the PyTorch function `nn.utils.parametrizations.spectral_norm`. We scale the inputs to the decoder to implement Lipschitz constraint scaling with $\lambda_S$.

**Reinforcement Learning** We use PPO (Schulman et al., 2017) for policy learning from rewards. For some domains, we use PPOBC (Jena et al., 2021). PPOBC augments the policy objective with a behavior cloning (BC) loss term which improves learning stability without directly affecting the discriminator or decoder. We highlight that using the BC term comes at no additional human cost, as demonstrations are already available in the IL setting. Furthermore, we make no assumptions about the demonstrations' behavior factors either and use the decoder network to infer the latent factor.

C.3 Domain-specific variations and tuning

The hyperparameters used in our optimization are listed in Tables 1, 2. Each method is independently tuned for $\lambda_I$ (and $\lambda_C$ for Lipz, GSD) over the specified ranges, to maximize MAE over the test split for K=10 over averaged over four rounds of evaluation and five train seeds. All hyperparameters omitted from the tables are set to default values from our base implementation.

| Hyperparameter | Value |
|---|---|
| NN update minibatch size | 256 |
| Policy learning rate | 3e-4 |
| Entropy bonus | 0.0001 |
| Gamma | 0.99 |
| GAE coefficient (Schulman et al., 2015) | 0.97 |
| NN architectures | FCN |
| Policy activation | Tanh |
| BC warmstart epochs | 10000 |
| Disc. activation | Tanh |
| Disc. learning rate | 1e-3 |
| Disc. gradient steps | 5 |
| Dec. hidden dimensions | [100, 256] |
| Dec. activation | ReLU |
| Dec. gradient steps | 5 |
| $\lambda_S$ | 10 |
| $b$ initial value | -5 |
| Constraint slack ($\epsilon$) | 1e-6 |
| Lambda learning rate | 1e-3 |
| Optimizers | Adam |

Table 1: The table contains the list of hyperparameters, common across domains and generalization settings. GAE: Generalized Advantage Estimation, NN: Neural Network, FCN: Fully connected network, BC: Behavior cloning, Disc.: Discriminator, Dec.: Decoder

| Hyperparam. \ Domain | HalfCheetah | DriveLaneshift | FetchPickPlace |
|---|---|---|---|
| Env. steps | 1.5e7 | 0.5e7 | 1e7 |
| RL algorithm | PPO | PPOBC | PPOBC |
| BC halflife, weight (PPOBC) | - | 0.1, 0.2 | 0.1, 0.1 |
| NN update interval (steps) | 10000 | 1000 | 10000 |
| NN hidden dimensions | [100, 100] | [100, 100] | [32, 32] |
| Observation norm. (w/ demos.) | False | False | True |
| Policy weight decay | (0, 1e-3) | 1e-4 | (1e-4, 5e-5) |
| Dec. learning rate | 1e-3 | 1e-4 | 1e-3 |
| Lambda initial value | (100, 500) | 500 | 5000 |
| Dec. gradient norm clip | $\infty$ | 25 | $\infty$ |
| Dec. rewards clip | [$-\infty, \infty$] | [-20, 5] | [$-\infty, \infty$] |
| Distillation objective weight | (0.02, 0.001) | (0.001, 0.001) | (0.0001, 0.0005) |
| $\lambda_S$ sweep list | [0.02, 0.05, 0.1, 0.2] | [0.1, 0.5, 1.0, 5.0] | [0.1, 0.5, 1.0, 5.0] |
| $\lambda_I$ sweep list | ([0.9, 0.8], [0.8, 0.7]) | ([0.99, 0.97], [0.99, 0.97]) | ([0.9, 0.8], [0.8, 0.7]) |

Table 2: The table contains the list of hyperparameters, specific to each domain (indicated by the column) and generalization setting (indicated by a 2-tuple (left, right), where left and right correspond to interpolation and extrapolation respectively).

## D    COMPARISON AGAINST OFFLINE IL APPROACHES

We compare GSD against offline IL approaches that learn solely from data without any environment interaction. We use implementations open-sourced by D3IL (Jia et al., 2024). We focus on multimodal action distribution modeling by excluding architectures that incorporate state histories or predict action sequences, such as action chunking (Zhao et al., 2023). We use fully connected neural network backbones with two hidden layers each containing 100 units. Unless otherwise specified, we use default hyperparameters that are most common across tasks in D3IL. We use demonstrations corresponding to held-out factor intervals as the validation dataset for early stopping.

- Behavior cloning (BC): Policy NN takes state as input and outputs actions, and is trained with a mean squared error (MSE) loss.
- BC with VAE (BC-VAE): BC-VAE uses a state-conditioned encoder-decoder setup to model the action distribution (Sohn et al., 2015). Latent space dimension is set to size 2, same as our approach. A weight of 5.0 is used for scaling KL-divergence loss.
- Implicit BC (IBC): Florence et al. (2022) propose energy models that implicitly capture the action distribution at each state. The action is inferred by optimizing the energy function using Markov chain Monte-carlo sampling at each inference step.
- K-Means Discretization (BeT): Shafiullah et al. (2022) propose an approach to capture multimodal action distributions using a learned discretization with $K$ predicted action means and offsets. $K$ is set to 64.
- Diffusion Policy (Diffusion): Chi et al. (2023); Pearce et al. (2023) propose modeling action distributions with a diffusion model conditioned on the state. Timestep embeddings of size 16 and 24 denoising steps are used.

We show the recovery and performance trade-off of offline IL and our approach in Fig. 12. With regard to task performance (x-axis), for HalfCheetah (top row) and FetchPickPlace (bottom), GSD outperforms offline approaches (except for interpolation in HalfCheetah) indicated by the red cross being positioned further right than others. For DriveLaneShift (middle row), offline approaches other than BC and BC-VAE are competitive with GSD. The result suggests that in domains with complex dynamics like HalfCheetah and FetchPickPlace, environment interaction is necessary for task completion when learning from few demonstrations.

With regard to recovery performance (y-axis), for HalfCheetah (top row) and FetchPickPlace (bottom), GSD outperforms all offline approaches, indicated by the red cross being positioned below others. For interpolation in DriveLaneShift (middle row, left), approaches IBC, BeT and Diffusion are comparable to GSD. For extrapolation (middle row, right), GSD outperforms all methods. Poor performance of offline approaches in domains with complex dynamics like HalfCheetah and FetchPickPlace may be attributed to the absence of environment interaction. In simpler domains, IBC, BeT or Diffusion may be able to interpolate diverse behaviors. However, for extrapolation, environment interaction is necessary, even for simpler domains.

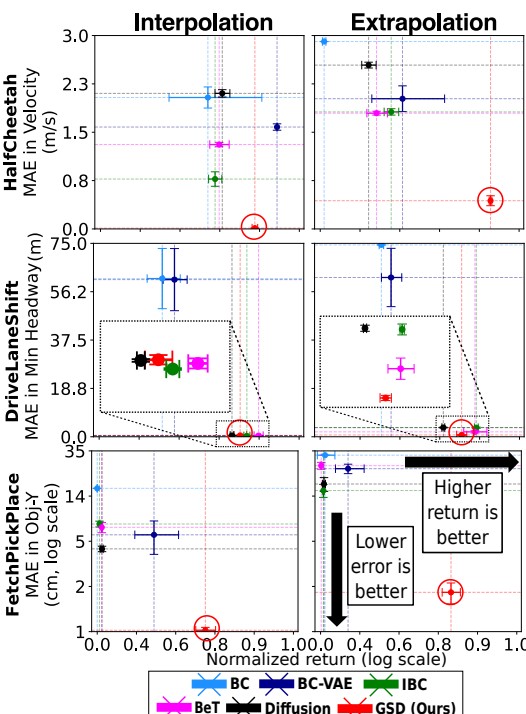

Figure 12: The figure shows the tradeoff between task and recovery performance for three domains and two splits for offline IL approaches and GSD (indicated with red circles). Error bars show standard errors over five seeds.

## E    EVALUATION WITH HUMAN DEMONSTRATIONS IN TABLE TENNIS

### E.1    SETUP

**Demonstrations:** The WAM arm is enabled with gravity compensation for collecting kinesthetic demonstrations. Messages published to a ROS interface are collected for two seconds (starting after the ball is detected to move over the table) which is enough time to capture the return trajectory. Joint states ($\mathbb{R}^7$) and ball positions ($\mathbb{R}^3$) are matched over time, and concatenated to construct state vectors ($\mathbb{R}^{10}$). Action vectors are constructed by calculating the displacements at corresponding timesteps. We collect five demonstrations for each of the two-stroke types considered.

**Simulation:** We use position control for the WAM arm at 100hz with the control gains tuned to match real robot demonstrations visually when replaying action commands open loop. We tune ball flight parameters such that the ball flight paths (before being struck) visually match those from the real demonstrations when launched from a similar position, velocity, and noise as the ball launcher. We add Gaussian noise to the ball positions in the observation vector to mimic real recorded ball positions. We use an episode length of 200 steps that corresponds to a real-life execution period of two seconds.

### E.2    METRICS

We detect if the ball has been returned by checking if the velocity component along the long edge of the TT table has reversed. Once a returning ball is detected, we check if the ball remains above the table plane and within 10 cm beyond the sides of the table for the following 0.5 seconds. If the trajectory satisfies both criteria, we deem it a success. We calculate the factor value for each successful return which is the maximum height the ball reaches in the return trajectory.

We sample five trajectories per sampled latent vector due to the stochastic nature of the ball observations. However, not every sampled trajectory for a particular latent vector is guaranteed to succeed due to the stochasticity in the domain and optimization. Thus, we consider a latent vector successful if the ball is returned in at least three out of five trials. We consider the factor value for the successful latent vector to be the mean of the values of the successful trajectories.

For each method and train seed, we sample 200 latent vectors from the prior, and sample five trajectories per latent vector. We report the fraction of successful latent vectors to evaluate if behaviors can accomplish the underlying task. Among the set of latent vectors, we subsample 100 successful latent vectors (after ensuring each method has at least 50% success rate) and report the entropy (based on particle estimates (Singh et al., 2003)) among the calculated factors using the equation shown in Eq. 8 where $V = \{v_i\}_{i=0}^M$ is the set of factor values $v_i$, $M = 100$, $K = 50$ and $\mathrm{NNe}_{K,V}(v_i)$ returns the $K$th nearest neighbor to $v_i$ from the set of values $V$. The entropy measure is up to a proportionality constant, as we use it to compare diversity achieved in return ball trajectory heights across methods.

$$H_K(V) = \frac{1}{M} \sum_{i=0}^M \log \|v_i - \mathrm{NNe}_{K,V}(v_i)\| \tag{8}$$

## F    FURTHER LIMITATIONS AND FUTURE WORK

Our approach requires a task-relevance measure, $f$, which we derived from demonstration occupancies. IL approaches that do not explicitly model expert occupancy (Reddy et al., 2020; Garg et al., 2021) may not be readily compatible for integration with our regularization. However, $Q$ functions learned during policy optimization may act as suitable alternatives for $f$.

Our regularization is implemented through approximately enforced constraints using Lagrange multipliers. Approximate enforcement may permit spurious behaviors that drastically vary from demonstrations. Parametric approaches akin to spectral normalization for Lipschitz continuity (Miyato et al., 2018) are desirable.

Instead of using diversity objectives, approaches that learn generalizable reward functions (Szot et al., 2023) could also be explored in the context of generalizable heterogeneous IL.

## G  GENERALITY OF $f$-RELEVANT DIVERSITY

Our $f$-relevant diversity formulation discussed in Sec. 5.1 is designed to encourage behavior diversity with respect to any defined energy measure $f$. We briefly demonstrate the generality of our formulation in the simple 2D PointMaze domain with a user-defined energy function as shown in Fig. 13. Our formulation has a potential application in diverse solution discovery (Kumar et al., 2020; Osa et al., 2022), where a bounded form of the estimated $Q$ function can be used as $f$ to encourage diversity in high value regions.

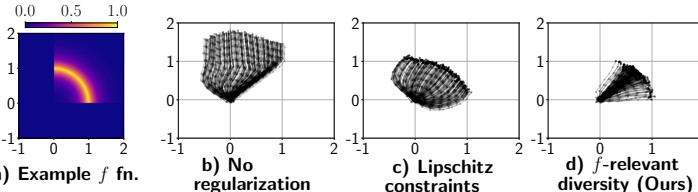

a) Example $f$ fn.   b) No regularization   c) Lipschitz constraints   d) $f$-relevant diversity (Ours)

Figure 13: The figure visualizes behaviors in 2D PointMaze learned with a predefined energy function $f$ shown in 13a. $f$ is used to specify rewards in 13b-d and additionally formulate our diversity objective in 13d. With no regularization (13b) or Lipschitz constraints (13c), trajectories visit low $f$-energy regions of the statespace. However, with $f$-relevant diversity (ours, 13d), a larger portion of trajectories cover diverse high energy states.

## H  EVALUATION RESULTS

We provide the numerical figures for recovery errors used to plot the graphs in Fig. 4 below. We further abbreviate Con, ConDist and Lipz as CO, CD and LZ respectively due to width constraints.

**HalfCheetah: Interpolation, Average:**

| Model | $K = 10$ | $K = 20$ | $K = 30$ | $K = 40$ | $K = 50$ |
|---|---|---|---|---|---|
| IG | $0.258 \pm 0.007$ | $0.147 \pm 0.006$ | $0.106 \pm 0.004$ | $0.080 \pm 0.004$ | $0.063 \pm 0.004$ |
| IG+LZ | $0.265 \pm 0.009$ | $0.156 \pm 0.010$ | $0.111 \pm 0.008$ | $0.089 \pm 0.008$ | $0.069 \pm 0.006$ |
| IG+CO | $0.229 \pm 0.015$ | $0.114 \pm 0.007$ | $0.074 \pm 0.004$ | $0.058 \pm 0.003$ | $0.049 \pm 0.003$ |
| IG+CD | $0.266 \pm 0.014$ | $0.142 \pm 0.007$ | $0.101 \pm 0.006$ | $0.077 \pm 0.005$ | $0.060 \pm 0.005$ |
| IG+CD+LZ | $0.307 \pm 0.004$ | $0.177 \pm 0.005$ | $0.124 \pm 0.004$ | $0.091 \pm 0.003$ | $0.077 \pm 0.003$ |
| GSD (Ours) | $0.226 \pm 0.007$ | $0.120 \pm 0.003$ | $0.081 \pm 0.003$ | $0.065 \pm 0.001$ | $0.052 \pm 0.002$ |

**HalfCheetah: Interpolation, Worst:**

| Model | $K = 10$ | $K = 20$ | $K = 30$ | $K = 40$ | $K = 50$ |
|---|---|---|---|---|---|
| IG | $0.298 \pm 0.004$ | $0.177 \pm 0.002$ | $0.134 \pm 0.003$ | $0.103 \pm 0.003$ | $0.076 \pm 0.003$ |
| IG+LZ | $0.348 \pm 0.020$ | $0.212 \pm 0.021$ | $0.159 \pm 0.019$ | $0.131 \pm 0.017$ | $0.099 \pm 0.013$ |
| IG+CO | $0.291 \pm 0.029$ | $0.142 \pm 0.014$ | $0.092 \pm 0.007$ | $0.072 \pm 0.007$ | $0.061 \pm 0.005$ |
| IG+CD | $0.339 \pm 0.024$ | $0.181 \pm 0.013$ | $0.133 \pm 0.011$ | $0.102 \pm 0.009$ | $0.082 \pm 0.008$ |
| IG+CD+LZ | $0.373 \pm 0.007$ | $0.223 \pm 0.010$ | $0.158 \pm 0.008$ | $0.117 \pm 0.006$ | $0.100 \pm 0.006$ |
| GSD (Ours) | $0.287 \pm 0.007$ | $0.151 \pm 0.005$ | $0.106 \pm 0.004$ | $0.084 \pm 0.002$ | $0.069 \pm 0.002$ |

**HalfCheetah: Extrapolation, Average:**

| Model | $K = 10$ | $K = 20$ | $K = 30$ | $K = 40$ | $K = 50$ |
|---|---|---|---|---|---|
| IG | $0.841 \pm 0.010$ | $0.766 \pm 0.014$ | $0.737 \pm 0.016$ | $0.710 \pm 0.017$ | $0.697 \pm 0.019$ |
| IG+LZ | $0.891 \pm 0.004$ | $0.829 \pm 0.005$ | $0.804 \pm 0.005$ | $0.788 \pm 0.005$ | $0.779 \pm 0.005$ |
| IG+CO | $0.741 \pm 0.007$ | $0.618 \pm 0.002$ | $0.560 \pm 0.004$ | $0.525 \pm 0.006$ | $0.505 \pm 0.007$ |
| IG+CD | $0.801 \pm 0.024$ | $0.642 \pm 0.020$ | $0.581 \pm 0.019$ | $0.538 \pm 0.019$ | $0.513 \pm 0.020$ |
| IG+CD+LZ | $0.686 \pm 0.008$ | $0.597 \pm 0.008$ | $0.559 \pm 0.010$ | $0.539 \pm 0.010$ | $0.525 \pm 0.011$ |
| GSD (Ours) | $0.682 \pm 0.028$ | $0.556 \pm 0.032$ | $0.512 \pm 0.033$ | $0.484 \pm 0.034$ | $0.472 \pm 0.034$ |

**HalfCheetah: Extrapolation, Worst:**

| Model | $K = 10$ | $K = 20$ | $K = 30$ | $K = 40$ | $K = 50$ |
|---|---|---|---|---|---|
| IG | $0.910 \pm 0.013$ | $0.854 \pm 0.016$ | $0.837 \pm 0.017$ | $0.828 \pm 0.017$ | $0.821 \pm 0.017$ |
| IG+LZ | $0.999 \pm 0.004$ | $0.909 \pm 0.007$ | $0.880 \pm 0.009$ | $0.869 \pm 0.009$ | $0.860 \pm 0.010$ |
| IG+CO | $0.875 \pm 0.019$ | $0.779 \pm 0.021$ | $0.725 \pm 0.024$ | $0.685 \pm 0.027$ | $0.661 \pm 0.030$ |
| IG+CD | $0.963 \pm 0.031$ | $0.750 \pm 0.017$ | $0.686 \pm 0.012$ | $0.656 \pm 0.008$ | $0.634 \pm 0.007$ |
| IG+CD+LZ | $0.797 \pm 0.012$ | $0.719 \pm 0.011$ | $0.699 \pm 0.011$ | $0.689 \pm 0.011$ | $0.682 \pm 0.011$ |
| GSD (Ours) | $0.808 \pm 0.039$ | $0.679 \pm 0.038$ | $0.646 \pm 0.040$ | $0.632 \pm 0.041$ | $0.621 \pm 0.041$ |

**DriveLaneshift: Interpolation, Average:**

| Model | $K = 10$ | $K = 20$ | $K = 30$ | $K = 40$ | $K = 50$ |
|---|---|---|---|---|---|
| IG | $3.435 \pm 0.148$ | $2.305 \pm 0.132$ | $1.808 \pm 0.122$ | $1.489 \pm 0.105$ | $1.277 \pm 0.098$ |
| IG+LZ | $3.193 \pm 0.096$ | $2.101 \pm 0.097$ | $1.631 \pm 0.101$ | $1.345 \pm 0.078$ | $1.111 \pm 0.073$ |
| IG+CO | $5.246 \pm 0.500$ | $3.153 \pm 0.375$ | $2.377 \pm 0.333$ | $1.936 \pm 0.302$ | $1.712 \pm 0.278$ |
| IG+CD | $3.619 \pm 0.200$ | $2.246 \pm 0.151$ | $1.749 \pm 0.154$ | $1.419 \pm 0.163$ | $1.221 \pm 0.154$ |
| IG+CD+LZ | $2.785 \pm 0.196$ | $1.652 \pm 0.172$ | $1.302 \pm 0.156$ | $1.114 \pm 0.142$ | $0.985 \pm 0.128$ |
| GSD (Ours) | $2.343 \pm 0.134$ | $1.299 \pm 0.094$ | $0.877 \pm 0.061$ | $0.685 \pm 0.058$ | $0.530 \pm 0.045$ |

**DriveLaneshift: Interpolation, Worst:**

| Model | $K = 10$ | $K = 20$ | $K = 30$ | $K = 40$ | $K = 50$ |
|---|---|---|---|---|---|
| IG | $4.528 \pm 0.262$ | $3.294 \pm 0.226$ | $2.693 \pm 0.228$ | $2.280 \pm 0.193$ | $2.008 \pm 0.190$ |
| IG+LZ | $4.702 \pm 0.179$ | $3.315 \pm 0.187$ | $2.675 \pm 0.192$ | $2.253 \pm 0.152$ | $1.859 \pm 0.140$ |
| IG+CO | $6.843 \pm 0.690$ | $4.304 \pm 0.507$ | $3.329 \pm 0.441$ | $2.754 \pm 0.397$ | $2.469 \pm 0.380$ |
| IG+CD | $5.057 \pm 0.307$ | $3.329 \pm 0.249$ | $2.657 \pm 0.269$ | $2.171 \pm 0.282$ | $1.878 \pm 0.271$ |
| IG+CD+LZ | $3.895 \pm 0.314$ | $2.581 \pm 0.310$ | $2.079 \pm 0.291$ | $1.797 \pm 0.269$ | $1.637 \pm 0.241$ |
| GSD (Ours) | $3.009 \pm 0.246$ | $1.709 \pm 0.171$ | $1.138 \pm 0.104$ | $0.911 \pm 0.106$ | $0.687 \pm 0.079$ |

**DriveLaneshift: Extrapolation, Average:**

| Model | $K = 10$ | $K = 20$ | $K = 30$ | $K = 40$ | $K = 50$ |
|---|---|---|---|---|---|
| IG | $5.807 \pm 0.231$ | $4.889 \pm 0.269$ | $4.446 \pm 0.288$ | $4.144 \pm 0.297$ | $3.943 \pm 0.302$ |
| IG+LZ | $6.435 \pm 0.124$ | $5.692 \pm 0.193$ | $5.375 \pm 0.224$ | $5.134 \pm 0.244$ | $4.977 \pm 0.257$ |
| IG+CO | $6.347 \pm 0.769$ | $4.335 \pm 0.584$ | $3.588 \pm 0.567$ | $3.166 \pm 0.542$ | $2.967 \pm 0.521$ |
| IG+CD | $3.902 \pm 0.185$ | $2.480 \pm 0.185$ | $1.886 \pm 0.172$ | $1.559 \pm 0.166$ | $1.399 \pm 0.168$ |
| IG+CD+LZ | $5.588 \pm 0.511$ | $3.809 \pm 0.400$ | $2.902 \pm 0.369$ | $2.472 \pm 0.362$ | $2.206 \pm 0.369$ |
| GSD (Ours) | $2.803 \pm 0.108$ | $1.545 \pm 0.074$ | $1.019 \pm 0.053$ | $0.815 \pm 0.046$ | $0.695 \pm 0.048$ |

**DriveLaneshift: Extrapolation, Worst:**

| Model | $K = 10$ | $K = 20$ | $K = 30$ | $K = 40$ | $K = 50$ |
|---|---|---|---|---|---|
| IG | $6.386 \pm 0.219$ | $5.527 \pm 0.246$ | $5.102 \pm 0.268$ | $4.805 \pm 0.289$ | $4.624 \pm 0.291$ |
| IG+LZ | $6.908 \pm 0.125$ | $6.222 \pm 0.224$ | $6.071 \pm 0.244$ | $5.923 \pm 0.259$ | $5.836 \pm 0.269$ |
| IG+CO | $8.599 \pm 1.302$ | $6.673 \pm 1.140$ | $5.827 \pm 1.107$ | $5.324 \pm 1.069$ | $5.033 \pm 1.026$ |
| IG+CD | $5.326 \pm 0.338$ | $3.606 \pm 0.341$ | $2.907 \pm 0.339$ | $2.438 \pm 0.322$ | $2.206 \pm 0.328$ |
| IG+CD+LZ | $8.723 \pm 1.105$ | $6.473 \pm 0.890$ | $5.038 \pm 0.790$ | $4.340 \pm 0.764$ | $3.945 \pm 0.766$ |
| GSD (Ours) | $4.067 \pm 0.225$ | $2.283 \pm 0.156$ | $1.534 \pm 0.105$ | $1.238 \pm 0.091$ | $1.034 \pm 0.094$ |

**FetchPickPlace: Interpolation, Average:**

| Model | $K = 10$ | $K = 20$ | $K = 30$ | $K = 40$ | $K = 50$ |
|---|---|---|---|---|---|
| IG | $0.071 \pm 0.004$ | $0.045 \pm 0.004$ | $0.034 \pm 0.003$ | $0.027 \pm 0.003$ | $0.022 \pm 0.002$ |
| IG+LZ | $0.044 \pm 0.002$ | $0.024 \pm 0.001$ | $0.016 \pm 0.001$ | $0.012 \pm 0.001$ | $0.010 \pm 0.001$ |
| IG+CO | $0.072 \pm 0.009$ | $0.053 \pm 0.010$ | $0.046 \pm 0.010$ | $0.041 \pm 0.011$ | $0.040 \pm 0.011$ |
| IG+CD | $0.083 \pm 0.011$ | $0.067 \pm 0.012$ | $0.060 \pm 0.012$ | $0.055 \pm 0.012$ | $0.052 \pm 0.012$ |
| IG+CD+LZ | $0.050 \pm 0.001$ | $0.027 \pm 0.001$ | $0.019 \pm 0.001$ | $0.015 \pm 0.001$ | $0.012 \pm 0.001$ |
| GSD (Ours) | $0.037 \pm 0.001$ | $0.020 \pm 0.001$ | $0.014 \pm 0.000$ | $0.011 \pm 0.000$ | $0.008 \pm 0.000$ |

**FetchPickPlace: Interpolation, Worst:**

| Model | $K = 10$ | $K = 20$ | $K = 30$ | $K = 40$ | $K = 50$ |
|---|---|---|---|---|---|
| IG | $0.084 \pm 0.005$ | $0.058 \pm 0.005$ | $0.046 \pm 0.005$ | $0.037 \pm 0.004$ | $0.031 \pm 0.004$ |
| IG+LZ | $0.056 \pm 0.003$ | $0.032 \pm 0.002$ | $0.021 \pm 0.002$ | $0.016 \pm 0.001$ | $0.013 \pm 0.001$ |
| IG+CO | $0.081 \pm 0.009$ | $0.061 \pm 0.011$ | $0.054 \pm 0.011$ | $0.048 \pm 0.011$ | $0.047 \pm 0.011$ |
| IG+CD | $0.088 \pm 0.010$ | $0.072 \pm 0.012$ | $0.064 \pm 0.012$ | $0.059 \pm 0.012$ | $0.056 \pm 0.012$ |
| IG+CD+LZ | $0.053 \pm 0.002$ | $0.030 \pm 0.001$ | $0.021 \pm 0.001$ | $0.017 \pm 0.001$ | $0.014 \pm 0.001$ |
| GSD (Ours) | $0.042 \pm 0.001$ | $0.022 \pm 0.001$ | $0.015 \pm 0.001$ | $0.013 \pm 0.001$ | $0.009 \pm 0.000$ |

**FetchPickPlace: Extrapolation, Average:**

| Model | $K = 10$ | $K = 20$ | $K = 30$ | $K = 40$ | $K = 50$ |
|---|---|---|---|---|---|
| IG | $0.137 \pm 0.003$ | $0.103 \pm 0.003$ | $0.087 \pm 0.003$ | $0.075 \pm 0.003$ | $0.068 \pm 0.003$ |
| IG+LZ | $0.083 \pm 0.003$ | $0.057 \pm 0.003$ | $0.045 \pm 0.003$ | $0.038 \pm 0.003$ | $0.034 \pm 0.003$ |
| IG+CO | $0.210 \pm 0.016$ | $0.183 \pm 0.018$ | $0.167 \pm 0.018$ | $0.158 \pm 0.018$ | $0.152 \pm 0.018$ |
| IG+CD | $0.264 \pm 0.012$ | $0.246 \pm 0.013$ | $0.233 \pm 0.014$ | $0.225 \pm 0.014$ | $0.220 \pm 0.015$ |
| IG+CD+LZ | $0.078 \pm 0.003$ | $0.046 \pm 0.002$ | $0.032 \pm 0.002$ | $0.026 \pm 0.002$ | $0.022 \pm 0.002$ |
| GSD (Ours) | $0.068 \pm 0.002$ | $0.037 \pm 0.002$ | $0.027 \pm 0.002$ | $0.021 \pm 0.002$ | $0.018 \pm 0.002$ |

**FetchPickPlace: Extrapolation, Worst:**

| Model | $K = 10$ | $K = 20$ | $K = 30$ | $K = 40$ | $K = 50$ |
|---|---|---|---|---|---|
| IG | $0.182 \pm 0.004$ | $0.143 \pm 0.004$ | $0.120 \pm 0.003$ | $0.105 \pm 0.003$ | $0.097 \pm 0.003$ |
| IG+LZ | $0.097 \pm 0.003$ | $0.071 \pm 0.004$ | $0.057 \pm 0.004$ | $0.049 \pm 0.004$ | $0.044 \pm 0.004$ |
| IG+CO | $0.249 \pm 0.012$ | $0.223 \pm 0.015$ | $0.208 \pm 0.017$ | $0.196 \pm 0.017$ | $0.191 \pm 0.018$ |
| IG+CD | $0.303 \pm 0.003$ | $0.286 \pm 0.006$ | $0.275 \pm 0.008$ | $0.266 \pm 0.009$ | $0.260 \pm 0.010$ |
| IG+CD+LZ | $0.092 \pm 0.003$ | $0.057 \pm 0.003$ | $0.042 \pm 0.004$ | $0.035 \pm 0.004$ | $0.031 \pm 0.004$ |
| GSD (Ours) | $0.094 \pm 0.005$ | $0.056 \pm 0.005$ | $0.041 \pm 0.004$ | $0.033 \pm 0.003$ | $0.028 \pm 0.004$ |

