# OpenReview forum: "Generalized Behavior Learning from Diverse Demonstrations"
_ICLR.cc/2025/Conference — ICLR 2025 Poster_

### Official Review · Reviewer_4BCW · 2024-10-15

**Soundness:** 4
**Presentation:** 4
**Contribution:** 3
**Rating:** 6
**Confidence:** 4

**Summary:**

This paper presents a novel framework called Guided Strategy Discovery (GSD) for learning diverse behaviors from demonstrations, aiming to generalize these behaviors to unseen task variations. The authors introduce a task-relevance diversity objective that encourages behavior exploration of latent factors in demonstrations, outperforming baseline methods in continuous control benchmarks like robot control and virtual table tennis.

**Strengths:**

- The work introduces a new approach for behavior learning that focuses on task-relevant diversity. The proposed GSD framework creatively balances imitation with exploration of unseen behavior spaces.
- The empirical evaluation is robust, spanning multiple domains (robot control, driving, and manipulation) and showing consistent improvement over baselines, especially in behavior generalization. The metrics used, such as interpolation and extrapolation errors, are well-chosen to highlight the effectiveness of GSD in diverse and novel task variations.
- The paper is well-structured, with clear explanations of the problem, related work, and contributions. Figures and diagrams effectively communicate the GSD framework and its comparative performance.

**Weaknesses:**

- The task-relevance diversity formulation, while promising, may still need refinement for more complex behaviors in physical systems, particularly where task success is less clear or difficult to define in terms of energy regions.
- The writing is mostly clear, but some sections on the technical implementation of the diversity objective could benefit from further simplification for a broader audience.
- The state-action space assumes a continuously varying task relevance, and the experiments are primarily performed in environments with this feature. The proposed method may fail to scale to settings where adjacent state-action pairs have significantly different energy.

**Questions:**

- Some existing works aim to learn diverse task-accomplishing behaviors from demonstrations that generalize over latent preferences using mutual information-based methods [1, 2] and latent space structure design [3]. It may help to discuss these existing works to motivate the need for the proposed method. How to convince people of a need of GSD given these prior works targeting the same goal?
- The constraint shown in Eq.1 assumes linearity of the state-action space. The proposed method may fail to scale to settings where adjacent state-action pairs have significantly different energy. How generally does this assumption apply to different tasks and state action space design?
- As pointed out in the limitation section, the experiments are constrained to one-dimensional latent factors, limiting the evaluation of the method's scalability to more complex, higher-dimensional spaces. Please analyze if and how GSD may potentially fail when applied to higher-dimensional latent space. And if not, why not evaluate such environments?


[1] Li, C., Blaes, S., Kolev, P., Vlastelica, M., Frey, J. and Martius, G., 2023, May. Versatile skill control via self-supervised adversarial imitation of unlabeled mixed motions. In 2023 IEEE international conference on robotics and automation (ICRA) (pp. 2944-2950). IEEE.

[2] Peng, X.B., Guo, Y., Halper, L., Levine, S. and Fidler, S., 2022. Ase: Large-scale reusable adversarial skill embeddings for physically simulated characters. ACM Transactions On Graphics (TOG), 41(4), pp.1-17.

[3] Li, C., Stanger-Jones, E., Heim, S. and Kim, S., 2024. FLD: Fourier Latent Dynamics for Structured Motion Representation and Learning. arXiv preprint arXiv:2402.13820.

---

> ### Author Response · Authors · 2024-11-21
> **Response to Reviewer 4BCW Part (1/3)**
>
> We thank you for your time and valuable feedback. Please find our response to the concerns raised. We invite the reviewer to follow up with clarifications or additional questions as needed.
>
> > The writing is mostly clear, but some sections on the technical implementation of the diversity objective could benefit from further simplification for a broader audience.
>
> For Algorithm 1, we will simplify the presentation by adding comments and pseudo code and move the detailed version to the Appendix. We welcome suggestions on how to make our manuscript more accessible.
>
> > The task-relevance diversity formulation, while promising, may still need refinement for more complex behaviors in physical systems, particularly where task success is less clear or difficult to define in terms of energy regions.
>
> Our approach, which is based on GAIL[10], inherits the assumption from GAIL and related IRL approaches that occupying the same state-action regions that expert demonstrations occupy correlates with task success. Our approach, like GAIL, captures occupancy into an energy function (i.e., the discriminator). As the reviewer notes, the assumption may not hold in more complex settings with non-expert demonstrators or partially observable environments, where high occupancy may not always indicate task success. In these cases, using an energy function to model occupancy by naively taking observation and action as input may not lead to successful task completion. We acknowledge that lifting these assumptions—such as by incorporating a history of observations or using more advanced modeling techniques—could be essential in more complex scenarios. We plan to explore these avenues in future work.
>
> Alternatively, task success in more complex tasks can be framed in terms of rewards. Approaches like Soft Q-learning [1] or SAC [2], which view reinforcement learning as probabilistic inference, provide a framework for learning Q-functions from rewards. These Q-functions could serve as energy functions that govern task success and occupancy, offering a potential way to extend our diversity objectives.  If rewards can sufficiently capture the critical aspects of task performance [3], then the use of $Q$ functions as energy functions could provide a promising avenue for scaling our approach. We believe that the combination of reward-based RL and our task-relevant diversity formulation offers exciting directions for future work.
>
> > As pointed out in the limitation section, the experiments are constrained to one-dimensional latent factors, limiting the evaluation of the method's scalability to more complex, higher-dimensional spaces. Please analyze if and how GSD may potentially fail when applied to higher-dimensional latent space. And if not, why not evaluate such environments?
>
> Our evaluation shows that GSD outperforms all baselines even with one-dimensional latent factors, indicating clear shortcomings in prior work [11, 12]. While one-dimensional latent factors may sound simple, the single dimension of variance can manifest as widely different distributions, capturing the complexity of multiple discrete modes. For example, in the Half-Cheetah domain, varying an agent’s velocity induces distinct gait patterns, a challenging behavior to extrapolate.
>
> Quality diversity approaches [17], which seek diverse behaviors but assume the availability of task performance metric and measure functions that provide factor values from state-action trajectories, typically also evaluate with only a few factors. In our setting where task performance metrics and measure functions are absent during learning, latent factors must be inferred from limited demonstrations and generalized to novel behaviors, while simultaneously ensuring task completion, even one dimensional factor present a significant challenge.
>
> We agree that evaluation with one-dimensional factors is a limitation. Our approach is not inherently restricted to one-dimensional factors, as the latent space dimension is a hyperparameter. We believe that GSD may scale well to more complex domains. However, scaling may depend on design choices unrelated to latent space regularization (e.g., methods to learn disentangled latent representations [15], transformers or other sequence models for modeling non-markovian latent factors [14], approaches for adversarial IL stability [13, 16]). We aim to do a full scalability analysis in future work. However, this study prioritizes validating GSD’s core contributions.

---

> ### Author Response · Authors · 2024-11-21
> **Response to Reviewer 4BCW Part (2/3)**
>
> > The state-action space assumes a continuously varying task relevance, and the experiments are primarily performed in environments with this feature. ... The constraint shown in Eq.1 assumes linearity of the state-action space. ... fail to scale to settings where adjacent state-action pairs have significantly different energy. How generally does this assumption apply to different tasks and state action space design?
>
> We thank the reviewer for their insightful question regarding applicability to settings where high-energy state-action pairs are non-adjacent and sparsely distributed. In the following paragraphs, we discuss how our diversity objective handles sparse distributions, potential shortcomings and measures that may help scaling to more complex tasks.
>
> Let us say that the agent is at a high-energy state-action pair, $<s, a>$, referred to as A hereon, associated with some demonstration $\tau$. Let $tau$ be associated with latent vector $z$. When the agent at A, using the policy $pi(.|.,z)$, transitions to neighboring low energy pairs, Eq. 1 enforces that all neighboring pairs are assigned the same latent vector $z$. Similar vector assignments in the neighborhood nullify the diversity incentive for other behaviors, say $pi(.|.,z’)$ to visit the neighborhood. While the diversity reward for $pi(.|.,z)$ to visit the neighborhood is also nullified, visitation of A is still ensured by the imitation reward $log(D(s, a, z’’))$. Similar reasoning applies to isolated high energy regions which are assigned their respective latent vectors. Their assignments are unconstrained, without any local structure, as high energy regions are non-adjacent.
>
> Lack of local structure in latent assignments begs the question: does the energy function, $f$, still generalize to novel regions beyond the demonstrations? We assume that the state-action space can be mapped to a connected low-dimensional feature space (manifold) because of continuous latent behavior factors. Mapping to a continuous feature space from disconnected high-energy regions must involve a complex function of the state description vector which energy functions like $f$, $g$, and the decoder $q$ must employ. Thus, scaling to more complex tasks may require additional strategies. For example, augmenting the architectures for $f$, $g$, and $q$, with conditioning approaches, such as attention mechanisms [4] or FiLM layers [5], or embedding functions that map state-action pairs to continuous feature spaces, could enhance generalization.
>
> Our assumption that the state-action space is mappable to a connected low-dimensional feature may be violated if the state-action space is not descriptive of the demonstrations’ variations. In such scenarios, it may not be possible to infer continuous latent factors. Including descriptors of variations into the state space or considering a history of states as input may be necessary. This is an important consideration, and we aim to explore these directions in future work to extend the scope of our method.
>
> Edit: We refined the text in this comment to ensure our discussion of the question is complete and accurate.

---

> ### Author Response · Authors · 2024-11-21
> **Response to Reviewer 4BCW Part (3/3)**
>
> > Some existing works aim to learn diverse task-accomplishing behaviors from demonstrations that generalize over latent preferences using mutual information-based methods [1, 2] and latent space structure design [3]. It may help to discuss these existing works to motivate the need for the proposed method. How to convince people of a need of GSD given these prior works targeting the same goal?
>
> CASSI [6] utilizes MI-based methods to learn diverse locomotion behaviors from a dataset of unlabeled motions that represent novel mixtures of individual motions. However, CASSI utilizes auxiliary task rewards to ensure that the novel behaviors result in successful locomotion without falling. In contrast, we demonstrate an ability to generate novel and successful behaviors without using auxiliary task rewards. We further evaluate our approach in a wider range of domains.
>
> FLD [7] learns to model diverse motions, focusing especially on exploiting temporal structure in periodic motion, by employing differentiable fast Fourier transforms of the learned latent space trajectories. FLD is able to achieve generalization over robot motion parameters. However, FLD is only applicable to periodic motions whereas our approach is generally applicable across domains.
>
> ASE [8] investigates the combination of diversity and imitation objectives to extract low level skills from unsegmented motion datasets. They explore sampling sequences of latent vectors, hyperspherical latent prior and regularization for the discriminator. Closely related work CALM [9] additionally uses conditioned discriminators to address mode collapse. However, both works omit regularization for the decoder and suffer from the same drawbacks as InfoGAIL (Sec. 4) which our work addresses.
>
> We will update our manuscript to include a discussion of these approaches to better motivate ours.
>
>
> [1] Haarnoja, T., Tang, H., Abbeel, P., & Levine, S. (2017, July). Reinforcement learning with deep energy-based policies. In the ICML (pp. 1352-1361). PMLR.
>
> [2] Haarnoja, T., Zhou, A., Abbeel, P., & Levine, S. (2018, July). Soft actor-critic: Off-policy maximum entropy deep reinforcement learning with a stochastic actor. In the ICML (pp. 1861-1870). PMLR.
>
> [3] Silver, D., Singh, S., Precup, D., & Sutton, R. S. (2021). Reward is enough. Artificial Intelligence, 299, 103535.
>
> [4] Vaswani, A. (2017). Attention is all you need. Advances in NeurIPS.
>
> [5] Perez, E., Strub, F., De Vries, H., Dumoulin, V., & Courville, A. (2018, April). Film: Visual reasoning with a general conditioning layer. In Proceedings of the AAAI conference on artificial intelligence (Vol. 32, No. 1).
>
> [6] Li, C., Blaes, S., Kolev, P., Vlastelica, M., Frey, J., & Martius, G. (2023, May). Versatile skill control via self-supervised adversarial imitation of unlabeled mixed motions. In 2023 IEEE international conference on robotics and automation (ICRA) (pp. 2944-2950). IEEE.
>
> [7] Li, C., Stanger-Jones, E., Heim, S., & bae Kim, S. FLD: Fourier Latent Dynamics for Structured Motion Representation and Learning. In The Twelfth ICLR.
>
> [8] Peng, X. B., Guo, Y., Halper, L., Levine, S., & Fidler, S. (2022). Ase: Large-scale reusable adversarial skill embeddings for physically simulated characters. ACM Transactions On Graphics (TOG), 41(4), 1-17.
>
> [9] Tessler, C., Kasten, Y., Guo, Y., Mannor, S., Chechik, G., & Peng, X. B. (2023, July). Calm: Conditional adversarial latent models for directable virtual characters. In ACM SIGGRAPH 2023 Conference Proceedings (pp. 1-9).
>
> [10] Ho, J., & Ermon, S. (2016). Generative adversarial imitation learning. Advances in NeurIPS, 29.
>
> [11] Li, Y., Song, J., & Ermon, S. (2017). Infogail: Interpretable imitation learning from visual demonstrations. Advances in NeurIPS, 30.
>
> [12] Park, S., Choi, J., Kim, J., Lee, H., & Kim, G. (2022). Lipschitz-constrained unsupervised skill discovery. In the ICLR.
>
> [13] Orsini, M., Raichuk, A., Hussenot, L., Vincent, D., Dadashi, R., Girgin, S., ... & Andrychowicz, M. (2021). What matters for adversarial imitation learning?. Advances in NeurIPS, 34, 14656-14668.
>
> [14] Vaswani, A. (2017). Attention is all you need. Advances in NeurIPS.
>
> [15] Mathieu, E., Rainforth, T., Siddharth, N., & Teh, Y. W. (2019, May). Disentangling disentanglement in variational autoencoders. In the ICML (pp. 4402-4412). PMLR.
>
> [16] Tessler, C., Kasten, Y., Guo, Y., Mannor, S., Chechik, G., & Peng, X. B. (2023, July). Calm: Conditional adversarial latent models for directable virtual characters. In ACM SIGGRAPH 2023 Conference Proceedings (pp. 1-9).
>
> [17] Batra, S., Tjanaka, B., Nikolaidis, S., & Sukhatme, G. (2024, July). Quality Diversity for Robot Learning: Limitations and Future Directions. In Proceedings of the Genetic and Evolutionary Computation Conference Companion (pp. 587-590).

---

> > ### Comment · Reviewer_4BCW · 2024-11-27
> > **thank you**
> >
> > Dear authors,
> >
> > Thanks for the effort in addressing my concerns and questions. However, as raised by other reviewers, the manuscript doesn't seem updated, and I couldn't find changes corresponding to the answers in the paper. It would be super helpful if you could update the manuscript accordingly.

---

> > > ### Author Response · Authors · 2024-11-28
> > > **Manuscript Updated**
> > >
> > > We have revised the manuscript based on all the reviewers’ feedback. Revisions are highlighted in red. We list the updates made in response to your concerns below:
> > > - Updated Algorithm 1 by adding descriptive sentences, equation references, citations, and removing excess notations. The detailed version is now in Appendix C.
> > > - Expanded limitations in Sec 7 (Lines 526-534) to discuss challenges with aspects related to scalability: high-dimensional or non-Markovian latent factors, sparse occupancy distributions, expert suboptimality, and partial observability.
> > > - Added discussion on structured approaches for heterogeneous IL [2, 3, 4, 5] in Sec 2 (Lines 132-139).

---

> > > ### Author Response · Authors · 2024-12-02
> > >
> > > We thank you for your valuable time and feedback. As the end of the rebuttal period is nearing, we remind and invite you to discuss any pending questions, concerns or feedback to further improve our manuscript.

---

### Official Review · Reviewer_F4MZ · 2024-11-03

**Soundness:** 3
**Presentation:** 3
**Contribution:** 3
**Rating:** 8
**Confidence:** 3

**Summary:**

This paper introduces Guided Strategy Discovery (GSD), which provides a new diversity formulation that is able to learn diverse, task-accomplishing behaviors from demonstrations. Through a 2d Point Maze task, the authors demonstrate that prior works fail to generalize to new behaviors. GSD introduces a new constraint $f$-relevant diversity to assign high energy to new state-action pairs. A learned task-relevance measure is used to guide the meaningful/task-relevant behavior patterns. The empirical experiments show that GSD outperforms other baselines in novel behavior discovery.

**Strengths:**

- The authors design a 2D toy task PointMaze, which demonstrates the existing diversity formulations, InfoGAIL, and Lipschitz constraints, fail to generate new, task-accomplishing behaviors.
- The authors provide an insightful motivation for the task-relevant diversity.
- The effectiveness of GSD is demonstrated on 3 simulation tasks and 1 real-to-sim Table Tennis task (The demonstrations are collected using a real-world setup).

**Weaknesses:**

- The experiments only contain tasks with 1-d latent factors, which restricts the applications of GSD. More complex task evaluations would greatly improve the quality of the paper.
- The experiments only consider the InfoGAIL setting. Extensions to other general imitation learning frameworks would be appreciated.
- The implementation seems complex to me. It would be great to provide the code for better understanding and reproduction.

**Questions:**

Overall, this paper provides the potential to learn a diverse range of novel behaviors while maintaining the task success rate. However, scaling to tasks with complex behaviors needs to be further evaluated. I have the following questions:

- In the experiments, the 1-d latent factors generally refer to position, velocity, or ball height. How would the authors expect to solve more complex behaviors in robot manipulation tasks? D3IL [1] provides multiple tasks with diverse behaviors. Taking the pushing task for example, could GSD be able to address this task when interacting with objects frequently?
- Could the diversity objectives be used in general behavior cloning?
- Regarding the energy function $f$, how to the authors model it? Maybe I missed something, but how do we get the desired energy function $f(s,a)$

[1] Towards Diverse Behaviors: A Benchmark for Imitation Learning with Human Demonstrations, ICLR’24

---

> ### Author Response · Authors · 2024-11-21
> **Response to Reviewer F4MZ Part (1/3)**
>
> We thank you for your time and valuable feedback. Please find our response to the concerns raised. We invite the reviewer to follow up with clarifications or additional questions as needed.
>
> > The experiments only contain tasks with 1-d latent factors, which restricts the applications of GSD.
> > ... scaling to tasks with complex behaviors needs to be further evaluated.
> > ... 1-d latent factors generally refer to position, velocity, or ball height. How would the authors expect to solve more complex behaviors in robot manipulation tasks?
>
> Our evaluation shows that GSD outperforms all baselines even with one-dimensional latent factors, indicating clear shortcomings in prior work [14, 15]. While one-dimensional latent factors may sound simple, the single dimension of variance can manifest as widely different distributions, capturing the complexity of multiple discrete modes. For example, in the Half-Cheetah domain, varying an agent’s velocity induces distinct gait patterns, a challenging behavior to extrapolate.
>
> Quality diversity approaches [20], which seek diverse behaviors but assume the availability of task performance metric and measure functions that provide factor values from state-action trajectories, typically also evaluate with only a few factors. In our setting where task performance metrics and measure functions are absent during learning, latent factors must be inferred from limited demonstrations and generalized to novel behaviors, while simultaneously ensuring task completion, even one dimensional factor present a significant challenge.
>
> We agree that evaluation with one-dimensional factors is a limitation. Our approach is not inherently restricted to one-dimensional factors, as the latent space dimension is a hyperparameter. We believe that GSD may scale well to more complex domains. However, scaling may depend on design choices unrelated to latent space regularization (e.g., methods to learn disentangled latent representations [18], transformers or other sequence models for modeling non-markovian latent factors [17], approaches for adversarial IL stability [16, 19]). We aim to do a full scalability analysis in future work. However, this study prioritizes validating GSD’s core contributions.
>
> > The experiments only consider the InfoGAIL setting. Extensions to other general imitation learning frameworks would be appreciated.
>
> We consider InfoGAIL to be representative of online methods that combine imitation learning (IL) and diverse behavior discovery using mutual information (MI). Existing work on representing diverse multimodal data [1, 2, 3, 4] typically operates in offline settings without environment interaction, unlike our approach. For online IL, non-adversarial approaches focus on simple reward functions [5] or structuring policies for high dimensional input [6, 7]. None of them address capturing the multimodality in expert demonstrations, like our approach. Incorporating expressive generative modeling techniques from offline IL into online multimodal IL is an exciting avenue for future work.
>
> We agree that including comparisons to offline approaches would provide a more comprehensive evaluation. We aim to provide these results in the coming few days -- please stay tuned.
>
> > The implementation seems complex to me. It would be great to provide the code for better understanding and reproduction.
>
> For Algorithm 1, we will simplify the presentation by adding comments and pseudo code and move the detailed version to the Appendix. We will make the code for our approach, including its variants, domains, and demonstrations, publicly available upon publication.
>
>
> > D3IL provides multiple tasks with diverse behaviors. Taking the pushing task for example, could GSD be able to address this task when interacting with objects frequently?
>
> Benchmarks for diverse behaviors, such as D3IL [8], typically feature tasks with discrete behavioral modalities. For example, in the D3IL pushing task, the four ways of completing the task -- pushing specific blocks into target zones-- are discrete, with no meaningful interpolation between them that still accomplishes the task. We expect our approach to be able to capture these discrete modes. However, such setups are unsuitable for evaluating the discovery of novel, continuous behavior variations, which is our primary objective.
>
> In contrast, our tasks (Sec 6.2) are designed to exhibit continuously varying factors with clear task objectives, enabling generalization from limited demonstrations and precise evaluation. Additionally, in the table tennis domain (Sec. 6.3), we use demonstrations of two different stroke types, where continuous generalization between them is meaningful, as captured by varying ball heights. These considerations ensure our evaluation setup is well-suited to assess our approach.

---

> > ### Author Response · Authors · 2024-11-26
> > **Comparison against offline IL approaches**
> >
> > We have updated the manuscript to include Section C3 to compare against offline IL methods that do not interact with the environment. We compare GSD against five approaches: Behavior Cloning (BC), BC with VAEs to model action distribution using a latent variable [1], Implicit Behavior Cloning (IBC) [2], Behavior Transformers [3] and Diffusion Policies [4, 5]. We use implementations open sourced by D3IL [6]. We focus our analysis on action modeling techniques by disregarding transformer backbones, access to state history or action-sequence prediction design [7]. We mostly use hyperparameters common across tasks in D3IL. We summarize our results below:
> > - In domains with complex dynamics, such as HalfCheetah and FetchPickPlace, GSD outperforms offline approaches in terms of both task and recovery performances. We attribute the result to the low number of demonstrations available for policy learning.
> > - In domains with simpler dynamics such as DriveLaneShift, GSD is comparable to IBC, BeT, Diffusion for interpolation in terms of both task and recovery performance. However, for extrapolation, GSD outperforms all approaches in recovery performance. The result highlights that online interaction may be necessary for extrapolating beyond training demonstrations. However, the competitive interpolation performance of these approaches in simpler domains highlights exciting avenues for their integration with diversity objectives in future work.
> >
> > [1] Sohn, K., Lee, H., & Yan, X. (2015). Learning structured output representation using deep conditional generative models. Advances in neural information processing systems, 28.
> >
> > [2] Florence, P., Lynch, C., Zeng, A., Ramirez, O. A., Wahid, A., Downs, L., ... & Tompson, J. (2022, January). Implicit behavioral cloning. In Conference on Robot Learning (pp. 158-168). PMLR.
> >
> > [3] Shafiullah, N. M., Cui, Z., Altanzaya, A. A., & Pinto, L. (2022). Behavior transformers: Cloning $ k $ modes with one stone. Advances in neural information processing systems, 35, 22955-22968.
> >
> > [4] Pearce, T., Rashid, T., Kanervisto, A., Bignell, D., Sun, M., Georgescu, R., ... & Devlin, S. (2023). Imitating human behaviour with diffusion models. arXiv preprint arXiv:2301.10677.
> >
> > [5] Chi, C., Xu, Z., Feng, S., Cousineau, E., Du, Y., Burchfiel, B., ... & Song, S. (2023). Diffusion policy: Visuomotor policy learning via action diffusion. The International Journal of Robotics Research, 02783649241273668.
> >
> > [6] Jia, X., Blessing, D., Jiang, X., Reuss, M., Donat, A., Lioutikov, R., & Neumann, G. (2024). Towards diverse behaviors: A benchmark for imitation learning with human demonstrations. arXiv preprint arXiv:2402.14606.
> >
> > [7] Zhao, T. Z., Kumar, V., Levine, S., & Finn, C. (2023). Learning fine-grained bimanual manipulation with low-cost hardware. arXiv preprint arXiv:2304.13705.

---

> > > ### Comment · Reviewer_F4MZ · 2024-11-26
> > >
> > > Thanks for your detailed response and additional experiments. I will raise my score to 8.

---

> > > > ### Author Response · Authors · 2024-11-28
> > > > **Manuscript Updated**
> > > >
> > > > Thank you for raising the score! We have revised the manuscript based on all the reviewers’ feedback. Revisions are highlighted in red. We list the updates made in response to your concerns below:
> > > > - Expanded limitations in Sec 7 (Lines 526-534) to discuss challenges with aspects related to scalability: high-dimensional or non-Markovian latent factors, sparse occupancy distributions, expert suboptimality, and partial observability.
> > > > - Explained the choice of InfoGAIL as the base method, alternative methods, and our focus on latent space modeling in Sec 6 (Lines 341-347).
> > > > - Included comparisons with offline baselines in Appendix D3.
> > > > - Updated Algorithm 1 by adding descriptive sentences, equation references, citations, and removing excess notations. The detailed version is now in Appendix C.
> > > > - Explained the unsuitability of D3IL for evaluating generalization over continuous behavior factors in Sec 6 (Lines 332-334).
> > > > - Updated the introduction of Sec 5.2 (Line 262) to clarify its purpose of deriving the energy function $f$ and improved flow with some edits.

---

> ### Author Response · Authors · 2024-11-21
> **Response to Reviewer F4MZ Part (2/3)**
>
> > Could the diversity objectives be used in general behavior cloning?
>
> Yes, but with a caveat. Applying diversity objectives like mutual information (MI) to behavior cloning (BC) may not be straightforward due MI’s dependence on states and actions. However, our regularization methodology can be applied to BC that uses variational autoencoders (VAEs) [11] to model multimodal action distributions. Let us explain in more detail:
>
> MI-based diversity objectives are usually used to capture diversity in state-action space regions into a latent variable. In this setting, they apply to learning from interaction data, such as with online interaction in regular or unsupervised RL [9] or existing datasets in offline RL [10]. Offline inverse RL [13] may be compatible with MI but we are unsure of any existing work. Non-RL approaches like BC that capture demonstration diversity typically utilize an autoencoding objective [11] for actions, which is closely related to MI through variational inference. Decoder regularization, such as spectral normalization for Lipschitz continuity can be applied to autoencoding to obtain smoothly varying latent embeddings [12]. $f$-relevant regularization, similar to our formulation $f$-relevant diversity, may be performed with an appropriate energy function, $f$, by scaling the Lipschitz constraint, as done in Equation 1. Definition of the energy function would depend on the problem motivation. For example, let us say that only at certain states (crucial junctures in decision making), allowing multimodal action distributions is desirable. If $f$ indicates such states with high energy, then application of $f$-relevant regularization would discourage multimodal action distributions at low energy states.
>
> > Regarding the energy function, how do the authors model it? Maybe I missed something, but how do we get the desired energy function
>
> We derive the energy function in Section 5.2 and will update the subsection opening to clarify its purpose and improve the flow of the text. We want the energy function to indicate regions with high energy whose occupancy by the learning agent is favorable for two objectives: (1) task completion and (2) diversity along the latent dimensions that demonstrations exhibit variation.
>
> If one were to model and train the discriminator (i.e., an energy function) as GAIL does, i.e., $D(s, a)$, it would capture occupancy of training demonstrations. However, the discriminator would fail to extend beyond the training demonstrations to indicate regions corresponding to novel locations along the demonstrations’ latent dimensions.
>
> We choose to condition the discriminator on the latent variable, i.e., $D(s, a, z)$. This conditioning allows the discriminator to capture occupancy specific to different latent vectors. However, we still require an unconditioned energy function to initialize the $f$-relevant diversity objective discussed in Section 5.1. To obtain the unconditioned function, we decompose the discriminator into unconditioned and conditioned bounded components, $f(s, a)$, $g(s, a, z)$, and penalize the magnitude of $g$. The penalization objective (Equation 4) encourages information from $g(s, a, z)$ to be distilled into $f(s, a)$. The conditioned component, $g(s, a, z)$, encouraged to be as minimal as possible, captures only regions crucial to the specific latent vector. The unconditioned component, $f$, indicates regions important for task completion along with all regions corresponding to demonstrators’ latent dimensions. Thus, the learned energy function satisfies our two objectives.

---

> ### Author Response · Authors · 2024-11-21
> **Response to Reviewer F4MZ Part (3/3)**
>
> [1] Florence, P., Lynch, C., Zeng, A., Ramirez, O. A., Wahid, A., Downs, L., ... & Tompson, J. (2022, January). Implicit behavioral cloning. In Conference on Robot Learning (pp. 158-168). PMLR.
>
> [2] Chi, C., Xu, Z., Feng, S., Cousineau, E., Du, Y., Burchfiel, B., ... & Song, S. (2023). Diffusion policy: Visuomotor policy learning via action diffusion. The International Journal of Robotics Research, 02783649241273668.
>
> [3] Shafiullah, N. M., Cui, Z., Altanzaya, A. A., & Pinto, L. (2022). Behavior transformers: Cloning $ k $ modes with one stone. Advances in neural information processing systems, 35, 22955-22968.
>
> [4] Zhao, T. Z., Kumar, V., Levine, S., & Finn, C. (2023). Learning fine-grained bimanual manipulation with low-cost hardware. arXiv preprint arXiv:2304.13705.
>
> [5] Haldar, S., Mathur, V., Yarats, D., & Pinto, L. (2023, March). Watch and match: Supercharging imitation with regularized optimal transport. In Conference on Robot Learning (pp. 32-43). PMLR.
>
> [6] Haldar, S., Pari, J., Rai, A., & Pinto, L. (2023). Teach a robot to fish: Versatile imitation from one minute of demonstrations. arXiv preprint arXiv:2303.01497.
>
> [7] Kumar, S., Zamora, J., Hansen, N., Jangir, R., & Wang, X. (2023, March). Graph inverse reinforcement learning from diverse videos. In Conference on Robot Learning (pp. 55-66). PMLR.
>
> [8] Jia, X., Blessing, D., Jiang, X., Reuss, M., Donat, A., Lioutikov, R., & Neumann, G. Towards Diverse Behaviors: A Benchmark for Imitation Learning with Human Demonstrations. In The Twelfth International Conference on Learning Representations.
>
> [9] Eysenbach, B., Gupta, A., Ibarz, J., & Levine, S. Diversity is All You Need: Learning Skills without a Reward Function. In the International Conference on Learning Representations.
>
> [10] Osa, T., & Harada, T. Discovering Multiple Solutions from a Single Task in Offline Reinforcement Learning. In the Forty-first International Conference on Machine Learning.
>
> [11] Kingma, D. P. (2013). Auto-encoding variational bayes. arXiv preprint arXiv:1312.6114.
>
> [12] Barrett, B., Camuto, A., Willetts, M., & Rainforth, T. (2022, May). Certifiably robust variational autoencoders. In the International Conference on Artificial Intelligence and Statistics (pp. 3663-3683). PMLR.
>
> [13] Chan, A. J., & van der Schaar, M. (2021, January). Scalable Bayesian Inverse Reinforcement Learning. In the International Conference on Learning Representations.
>
> [14] Li, Y., Song, J., & Ermon, S. (2017). Infogail: Interpretable imitation learning from visual demonstrations. Advances in neural information processing systems, 30.
>
> [15] Park, S., Choi, J., Kim, J., Lee, H., & Kim, G. (2022). Lipschitz-constrained unsupervised skill discovery. In the International Conference on Learning Representations.
>
> [16] Orsini, M., Raichuk, A., Hussenot, L., Vincent, D., Dadashi, R., Girgin, S., ... & Andrychowicz, M. (2021). What matters for adversarial imitation learning?. Advances in Neural Information Processing Systems, 34, 14656-14668.
>
> [17] Vaswani, A. (2017). Attention is all you need. Advances in Neural Information Processing Systems.
>
> [18] Mathieu, E., Rainforth, T., Siddharth, N., & Teh, Y. W. (2019, May). Disentangling disentanglement in variational autoencoders. In the International conference on machine learning (pp. 4402-4412). PMLR.
>
> [19] Tessler, C., Kasten, Y., Guo, Y., Mannor, S., Chechik, G., & Peng, X. B. (2023, July). Calm: Conditional adversarial latent models for directable virtual characters. In ACM SIGGRAPH 2023 Conference Proceedings (pp. 1-9).
>
> [20] Batra, S., Tjanaka, B., Nikolaidis, S., & Sukhatme, G. (2024, July). Quality Diversity for Robot Learning: Limitations and Future Directions. In Proceedings of the Genetic and Evolutionary Computation Conference Companion (pp. 587-590).

---

### Official Review · Reviewer_LHbJ · 2024-11-03

**Soundness:** 2
**Presentation:** 2
**Contribution:** 2
**Rating:** 6
**Confidence:** 2

**Summary:**

The authors address the challenge of generating diverse behaviors that are useful for a wide range of tasks. They introduce a novel approach called *Guided Strategy Discovery (GSD)*, which features a diversity formulation that uses a learned task-relevance measure to prioritize behaviors that effectively explore modeled latent factors. The proposed method is  evaluated across multiple simulated tasks and in a task with real-world data.

**Strengths:**

The paper addresses a significant and relevant problem , contributing to our understanding of behavior diversity in various tasks. The use of real-world human data for evaluation is praiseworthy, as it enhances the applicability findings. Additionally, the paper features well-designed and illustrative figures that support the discussed concepts .

**Weaknesses:**

The paper builds upon InfoGAIL, an adversarial imitation learning method; however, it does not explore other adversarial imitation learning (AIL) methods or potential modifications that could enhance performance. A more comprehensive overview and comparison of relevant AIL techniques, such as those discussed in Orsini et al [1], would strengthen the contextualization of the proposed approach and highlight its novelty by discussing how different design choices influence learning.

Furthermore, the tasks considered in the paper do not align with established benchmarks. For example, recent work has introduced benchmarks for diverse behaviors, as highlighted by Jia et al[2]. Additionally, the paper lacks comparisons to other methods, including diffusion-based approaches that can effectively represent diverse, multimodal data.

The writing quality of the paper could be improved for clarity and conciseness. The term "preference" does not align with its common usage in related fields such as Reinforcement Learning from Human Feedback (RLHF) [3], which may lead to confusion among readers.

In terms of presentation, the caption for Figure 10 appears integrated into the surrounding text, making it difficult to read. Additionally, Algorithm 1 (Lines 270-287) is quite dense and may overwhelm readers. The notation used in Algorithm 1, Line 274, could also be confusing due to the presence of commas following terms (e.g., policy, $\pi$).

[1]Orsini, M., Raichuk, A., Hussenot, L., Vincent, D., Dadashi, R., Girgin, S., Geist, M., Bachem, O., Pietquin, O., & Andrychowicz, M. (2021). What Matters for Adversarial Imitation Learning? In Advances in Neural Information Processing Systems

[2]Jia, X., Blessing, D., Jiang, X., Reuss, M., Donat, A., Lioutikov, R., & Neumann, G. (2024). *Towards Diverse Behaviors: A Benchmark for Imitation Learning with Human Demonstrations*. In *Proceedings of the Twelfth International Conference on Learning Representations (ICLR)*

[3]Christiano, P. F., Leike, J., Brown, T., Martic, M., Legg, S., & Amodei, D. (2017). *Deep Reinforcement Learning from Human Preferences*. In *Advances in Neural Information Processing Systems*

**Questions:**

In Appendix C.2, the authors state that they used PPOBC for the DriveLaneshift and FetchPickPlace tasks. However, it is important to note that PPOBC is not a pure reinforcement learning (RL) algorithm, as it incorporates demonstrations. How would the proposed method differ if it were implemented using only PPO without the influence of demonstrations?

Additionally, could the authors clarify what is meant by diverse behaviors in the context of the HalfCheetah task? Specifically, how are different speeds relevant for tasks in the sense of improved task performance?

Finally, will the code be made publicly available? Furthermore, will the dataset of real-world human demonstrations also be accessible to the research community?

[1]Rohit Jena, Changliu Liu, and Katia Sycara. Augmenting gail with bc for sample efficient imitation learning. In Conference on Robot Learning, pp. 80–90. PMLR, 2021

---

> ### Author Response · Authors · 2024-11-21
> **Response to Reviewer LHbJ Part (1/3)**
>
> We thank you for your time and valuable feedback. Please find our response to the concerns raised. We invite the reviewer to follow up with clarifications or additional questions as needed.
>
> > The paper builds upon InfoGAIL … does not explore other adversarial imitation learning (AIL) methods or potential modifications … such as those discussed in Orsini et al … would strengthen the contextualization of the proposed approach and highlight its novelty.
>
> We thank the reviewer for the suggestions to better contextualize our work. We will update related work (Sec. 2) and evaluation (Sec. 6) to include a discussion of AIL approaches. We will emphasize that InfoGAIL is discussed as a representative baseline combining AIL with Mutual Information (MI) [1]. Our framework is flexible and compatible with any AIL approach that formulates an energy function (e.g., a discriminator) to model expert occupancy. Such approaches [14, 15, 16] are common in online imitation learning [2]. In addition, we hypothesize that $Q$-functions may also serve as appropriate energy functions due to their connections to energy-based models [19, 20]. If true, our method will also be compatible with non-adversarial IL approaches [17, 18, 10]. Nevertheless, we already incorporate several improvements from Osrini et al. [2] to ensure stable baseline performance across methods and domains, as detailed in Appendix C1. However, our primary contribution is a methodology to perform latent space regularization utilizing these energy functions.
>
> > Furthermore, the tasks considered in the paper do not align with established benchmarks. For example, recent work has introduced benchmarks for diverse behaviors, as highlighted by Jia et al.
>
> We will update Sec. 6 to clarify why we designed our evaluation setup and why we deviated from established benchmarks. Benchmarks for diverse behaviors, such as those highlighted by Jia et al. [3], typically feature tasks with discrete behavioral modalities. For example, in the D3IL pushing task, the four ways of completing the task -- pushing specific blocks into target zones-- are discrete, with no meaningful interpolation between them that still accomplishes the task.. Such setups are unsuitable for evaluating the discovery of novel, continuous behavior variations.
>
> In contrast, our tasks (Sec 6.2) are designed to exhibit continuously varying factors with clear task objectives, enabling generalization from limited demonstrations and precise evaluation. Additionally , in the table tennis domain (Sec. 6.3), we use demonstrations of two different stroke types, where continuous generalization between the two is meaningful, as captured by varying ball heights. These considerations ensure our evaluation setup is well-suited to assess our approach.
>
> > Additionally, the paper lacks comparisons to other methods, including diffusion-based approaches that can effectively represent diverse, multimodal data.
>
> Existing work on representing diverse multimodal data [4, 5, 6, 7] typically operates in offline settings without environment interaction, unlike our approach. For online IL, diffusion methods largely focus on modeling the discriminator [8, 9]. Non-diffusion, non-adversarial approaches focus on simple reward functions [10] or structuring policies for high dimensional input [11, 12]. None of them address capturing the multimodality in expert demonstrations, like our approach. Incorporating expressive generative modeling techniques, such as diffusion, for online multimodal IL is an exciting avenue for future work, and we appreciate the suggestion.
>
> We agree that including comparisons to offline approaches would provide a more comprehensive evaluation. We aim to provide these results in the coming few days -- please stay tuned.
>
> > The term "preference" does not align with its common usage in related fields such as Reinforcement Learning from Human Feedback (RLHF) [3], which may lead to confusion among readers.
>
> As suggested, we will replace the term “preference” with “behavior factor” to align with its common usage in related fields and avoid potential confusion.
>
> > The writing quality of the paper could be improved for clarity and conciseness. In terms of presentation, the caption for Figure 10 appears integrated into the surrounding text, making it difficult to read. Additionally, Algorithm 1 (Lines 270-287) is quite dense and may overwhelm readers. The notation used in Algorithm 1, Line 274, could also be confusing due to the presence of commas following terms (e.g., policy,).
>
> We appreciate the feedback and will make the suggested changes to improve clarity. For Algorithm 1, we will simplify the presentation by adding comments and pseudo code and move the detailed version to the Appendix.

---

> > ### Author Response · Authors · 2024-11-26
> > **Comparison against offline IL approaches**
> >
> > We have updated the manuscript to include Section C3 to compare against offline IL methods that do not interact with the environment. We compare GSD against five approaches: Behavior Cloning (BC), BC with VAEs to model action distribution using a latent variable [1], Implicit Behavior Cloning (IBC) [2], Behavior Transformers [3] and Diffusion Policies [4, 5]. We use implementations open sourced by D3IL [6]. We focus our analysis on action modeling techniques by disregarding transformer backbones, access to state history or action-sequence prediction design [7]. We mostly use hyperparameters common across tasks in D3IL. We summarize our results below:
> > - In domains with complex dynamics, such as HalfCheetah and FetchPickPlace, GSD outperforms offline approaches in terms of both task and recovery performances. We attribute the result to the low number of demonstrations available for policy learning.
> > - In domains with simpler dynamics such as DriveLaneShift, GSD is comparable to IBC, BeT, Diffusion for interpolation in terms of both task and recovery performance. However, for extrapolation, GSD outperforms all approaches in recovery performance. The result highlights that online interaction may be necessary for extrapolating beyond training demonstrations. However, the competitive interpolation performance of these approaches in simpler domains highlights exciting avenues for their integration with diversity objectives in future work.
> >
> > [1] Sohn, K., Lee, H., & Yan, X. (2015). Learning structured output representation using deep conditional generative models. Advances in neural information processing systems, 28.
> >
> > [2] Florence, P., Lynch, C., Zeng, A., Ramirez, O. A., Wahid, A., Downs, L., ... & Tompson, J. (2022, January). Implicit behavioral cloning. In Conference on Robot Learning (pp. 158-168). PMLR.
> >
> > [3] Shafiullah, N. M., Cui, Z., Altanzaya, A. A., & Pinto, L. (2022). Behavior transformers: Cloning $ k $ modes with one stone. Advances in neural information processing systems, 35, 22955-22968.
> >
> > [4] Pearce, T., Rashid, T., Kanervisto, A., Bignell, D., Sun, M., Georgescu, R., ... & Devlin, S. (2023). Imitating human behaviour with diffusion models. arXiv preprint arXiv:2301.10677.
> >
> > [5] Chi, C., Xu, Z., Feng, S., Cousineau, E., Du, Y., Burchfiel, B., ... & Song, S. (2023). Diffusion policy: Visuomotor policy learning via action diffusion. The International Journal of Robotics Research, 02783649241273668.
> >
> > [6] Jia, X., Blessing, D., Jiang, X., Reuss, M., Donat, A., Lioutikov, R., & Neumann, G. (2024). Towards diverse behaviors: A benchmark for imitation learning with human demonstrations. arXiv preprint arXiv:2402.14606.
> >
> > [7] Zhao, T. Z., Kumar, V., Levine, S., & Finn, C. (2023). Learning fine-grained bimanual manipulation with low-cost hardware. arXiv preprint arXiv:2304.13705.

---

> > > ### Comment · Reviewer_LHbJ · 2024-11-26
> > >
> > > Dear authors, thank you for your answers. Based on them, I would be inclined to increase the score. However, before that, I would appreciate it if you would provide the updated manuscript that contains all the mentioned changes and additions.

---

> > > > ### Author Response · Authors · 2024-11-28
> > > > **Manuscript Updated**
> > > >
> > > > We have revised the manuscript based on all the reviewers’ feedback. Revisions are highlighted in red. We list the updates made in response to your concerns below:
> > > > - Updated related work to discuss online and adversarial IL approaches in Sec 2 (Lines 114-120)
> > > > - Explained the choice of InfoGAIL as the base method, alternative methods, and our focus on latent space modeling in Sec 6 (Lines 341-347).
> > > > - Explained the unsuitability of D3IL for evaluating generalization over continuous behavior factors in Sec 6 (Lines 332-334).
> > > > - Included comparisons with offline baselines in Appendix D3.
> > > > - Replaced the term "preference" with "behavior factor" to avoid confusion with its common usage [1] and provided a definition in Sec 1 (Lines 57-58).
> > > > - Fixed formatting issues with Fig 10. Updated Algorithm 1 by adding descriptive sentences, equation references, citations, and removing excess notations. The detailed version is now in Appendix C.
> > > > - Clarified the role of the auxiliary BC objective used with PPO for certain domains in Appendix D (Lines 1180-1184).

---

> ### Author Response · Authors · 2024-11-21
> **Response to Reviewer LHbJ Part (2/3)**
>
> > In Appendix C.2, the authors state that they used PPOBC for the DriveLaneshift and FetchPickPlace tasks. However, it is important to note that PPOBC is not a pure reinforcement learning (RL) algorithm, as it incorporates demonstrations. How would the proposed method differ if it were implemented using only PPO without the influence of demonstrations?
>
> The task-relevant diversity objective, which forms our core contribution, is independent of the RL routine used to learn the policy. We choose PPOBC for DriveLaneshift and FetchPickPlace, as it stabilizes learning for these domains. PPOBC augments the policy objective with a behavior cloning (BC) loss term which does not affect the discriminator or decoder objectives. If the RL routine were implemented using only PPO (i.e., without a BC term), the approach would remain the same but the absence of the BC loss could lead to less stable learning. We want to emphasize that using the BC loss term comes at no additional human cost, as demonstrations are already available in the IL setting. Furthermore, we make no assumptions about the demonstrations’ behavior factors either and use the decoder network to infer the latent factor.
>
> > Additionally, could the authors clarify what is meant by diverse behaviors in the context of the HalfCheetah task? Specifically, how are different speeds relevant for tasks in the sense of improved task performance?
>
> In HalfCheetah, diverse behaviors refer to the robot’s ability to perform forward movement at varying speeds. The task to move forward is quantified by the number of timesteps in which the robot moves forward a non-zero amount, and all behaviors --regardless of speed-- receive high task rewards. Full details are provided in Appendix B1.
>
> > Finally, will the code be made publicly available? Furthermore, will the dataset of real-world human demonstrations also be accessible to the research community?
>
> Yes, we will make the code for our approach, including its variants, domains, and demonstrations, publicly available upon publication.

---

> ### Author Response · Authors · 2024-11-21
> **Response to Reviewer LHbJ Part (3/3)**
>
> [1] Eysenbach, B., Gupta, A., Ibarz, J., & Levine, S. Diversity is All You Need: Learning Skills without a Reward Function. In International Conference on Learning Representations.
>
> [2] Orsini, M., Raichuk, A., Hussenot, L., Vincent, D., Dadashi, R., Girgin, S., Geist, M., Bachem, O., Pietquin, O., & Andrychowicz, M. (2021). What Matters for Adversarial Imitation Learning? In Advances in Neural Information Processing Systems
>
> [3] Jia, X., Blessing, D., Jiang, X., Reuss, M., Donat, A., Lioutikov, R., & Neumann, G. Towards Diverse Behaviors: A Benchmark for Imitation Learning with Human Demonstrations. In The Twelfth International Conference on Learning Representations.
>
> [4] Florence, P., Lynch, C., Zeng, A., Ramirez, O. A., Wahid, A., Downs, L., ... & Tompson, J. (2022, January). Implicit behavioral cloning. In Conference on Robot Learning (pp. 158-168). PMLR.
>
> [5] Chi, C., Xu, Z., Feng, S., Cousineau, E., Du, Y., Burchfiel, B., ... & Song, S. (2023). Diffusion policy: Visuomotor policy learning via action diffusion. The International Journal of Robotics Research, 02783649241273668.
>
> [6] Shafiullah, N. M., Cui, Z., Altanzaya, A. A., & Pinto, L. (2022). Behavior transformers: Cloning $ k $ modes with one stone. Advances in neural information processing systems, 35, 22955-22968.
>
> [7] Zhao, T. Z., Kumar, V., Levine, S., & Finn, C. (2023). Learning fine-grained bimanual manipulation with low-cost hardware. arXiv preprint arXiv:2304.13705.
>
> [8] Wang, B., Wu, G., Pang, T., Zhang, Y., & Yin, Y. (2024, March). DiffAIL: Diffusion Adversarial Imitation Learning. In Proceedings of the AAAI Conference on Artificial Intelligence (Vol. 38, No. 14, pp. 15447-15455).
>
> [9] Lai, C. M., Wang, H. C., Hsieh, P. C., Wang, Y. C. F., Chen, M. H., & Sun, S. H. (2024). Diffusion-Reward Adversarial Imitation Learning. arXiv preprint arXiv:2405.16194.
>
> [10] Haldar, S., Mathur, V., Yarats, D., & Pinto, L. (2023, March). Watch and match: Supercharging imitation with regularized optimal transport. In Conference on Robot Learning (pp. 32-43). PMLR.
>
> [11] Haldar, S., Pari, J., Rai, A., & Pinto, L. (2023). Teach a robot to fish: Versatile imitation from one minute of demonstrations. arXiv preprint arXiv:2303.01497.
>
> [12] Kumar, S., Zamora, J., Hansen, N., Jangir, R., & Wang, X. (2023, March). Graph inverse reinforcement learning from diverse videos. In Conference on Robot Learning (pp. 55-66). PMLR.
>
> [13] Jena, R., Liu, C., & Sycara, K. (2021, October). Augmenting gail with bc for sample efficient imitation learning. In Conference on Robot Learning (pp. 80-90). PMLR.
>
> [14] Ho, J., & Ermon, S. (2016). Generative adversarial imitation learning. Advances in neural information processing systems, 29.
>
> [15] Fu, J., Luo, K., & Levine, S. (2017). Learning robust rewards with adversarial inverse reinforcement learning. arXiv preprint arXiv:1710.11248.
>
> [16] Kostrikov, I., Agrawal, K. K., Dwibedi, D., Levine, S., & Tompson, J. (2018). Discriminator-actor-critic: Addressing sample inefficiency and reward bias in adversarial imitation learning. arXiv preprint arXiv:1809.02925.
>
> [17] Reddy, S., Dragan, A. D., & Levine, S. (2019). Sqil: Imitation learning via reinforcement learning with sparse rewards. arXiv preprint arXiv:1905.11108.
>
> [18] Garg, D., Chakraborty, S., Cundy, C., Song, J., & Ermon, S. (2021). Iq-learn: Inverse soft-q learning for imitation. Advances in Neural Information Processing Systems, 34, 4028-4039.
>
> [19] Haarnoja, T., Tang, H., Abbeel, P., & Levine, S. (2017, July). Reinforcement learning with deep energy-based policies. In the International conference on machine learning (pp. 1352-1361). PMLR.
>
> [20] Haarnoja, T., Zhou, A., Abbeel, P., & Levine, S. (2018, July). Soft actor-critic: Off-policy maximum entropy deep reinforcement learning with a stochastic actor. In the International conference on machine learning (pp. 1861-1870). PMLR.

---

> ### Comment · Reviewer_LHbJ · 2024-11-28
>
> Thank you for updating the manuscript. As mentioned before, based on the updated manuscript as well as the provided answers, I will increase my score.

---

> > ### Author Response · Authors · 2024-12-02
> >
> > Thank you for your valuable time, feedback and raising the score! We welcome any pending questions, concerns or feedback to further improve our manuscript.

---

### Official Review · Reviewer_kDeW · 2024-11-04

**Soundness:** 3
**Presentation:** 2
**Contribution:** 2
**Rating:** 6
**Confidence:** 2

**Summary:**

This paper proposes a novel diversity formulation based on a learned task-relevance measure that prioritizes behaviors exploring modeled latent factors, called Guided Strategy Discovery (GSD).
The proposed method is empirically validated across three continuous control benchmarks for generalizing to in-distribution (interpolation) and out-of-distribution (extrapolation) preferences, as well as demonstrated to generalize striking behaviors for table tennis in a virtual testbed while leveraging human demonstrations collected in the real world.

**Strengths:**

The paper is written in a clear way and is well structured. The proposed method is formulated clearly and scoped within relevant literature. The proposed method is also evaluated experimentally, including comparisons with alternatives. Figures and visualisations are clear and help capture the contribution of the method.

**Weaknesses:**

Some results are not immediately clear from the plots, for example the visualisation of the interpolation/extrapolation experiments in Figure 5 is difficult to read.
It is unclear if the proposed method can scale up in terms, for example, of dimensions of preferences, tasks, rewards.

**Questions:**

What are the limiting factors of the chosen regularization in the latent space?
How much is the representation limited by demonstrations? Does this approach suffer from problems similar to behavior cloning?
Are there more intuitive visualisations for your results, eg for interpolation/extrapolation?
Is exploration limited and does this have an effect on final performance?

---

> ### Author Response · Authors · 2024-11-21
> **Response to Reviewer kDeW Part (1/3)**
>
> We thank you for your time and valuable feedback. Please find our response to the concerns raised. We invite the reviewer to follow up with clarifications or additional questions as needed.
>
> > Some results are not immediately clear from the plots... visualisation of the interpolation/extrapolation experiments in Figure 5 is difficult to read.
>
> We will improve the scatter plots in Figure 5 by adding perpendiculars to the axes, making the values easier to read and enabling direct comparisons. Additionally, we will include arrows to indicate the desired values for each axis, enhancing clarity and interpretability.
>
> > Are there more intuitive visualizations for your results, eg for interpolation/extrapolation?
>
> In Figure 6 with FetchPickPlace, we visualize the learned latent space by coloring cells with the achieved object placement location. Our approach can construct a well-structured latent space (Figure 6d) and discover novel behaviors (red, green, and dark blue cells) by utilizing demonstrations (yellow, light blue cells), without any access to the ground-truth factor values. We are open to suggestions on providing more intuitive visualizations.
>
> > It is unclear if the proposed method can scale up in terms, for example, of dimensions of preferences, tasks, rewards.
>
> Our evaluation shows that GSD outperforms all baselines even with one-dimensional latent factors, indicating clear shortcomings in prior work [11, 12]. While one-dimensional latent factors may sound simple, the single dimension of variance can manifest as widely different distributions, capturing the complexity of multiple discrete modes. For example, in the Half-Cheetah domain, varying an agent’s velocity induces distinct gait patterns, a challenging behavior to extrapolate.
>
> Quality diversity approaches [17], which seek diverse behaviors but assume the availability of task performance metric and measure functions that provide factor values from state-action trajectories, typically also evaluate with only a few factors. In our setting where task performance metrics and measure functions are absent during learning, latent factors must be inferred from limited demonstrations and generalized to novel behaviors, while simultaneously ensuring task completion, even one dimensional factor present a significant challenge.
>
> We agree that evaluation with one-dimensional factors is a limitation. Our approach is not inherently restricted to one-dimensional factors, as the latent space dimension is a hyperparameter. We believe that GSD may scale well to more complex domains. However, scaling may depend on design choices unrelated to latent space regularization (e.g., methods to learn disentangled latent representations [15], transformers or other sequence models for modeling non-markovian latent factors [14], approaches for adversarial IL stability [13, 16]). We aim to do a full scalability analysis in future work. However, this study prioritizes validating GSD’s core contributions.

---

> ### Author Response · Authors · 2024-11-21
> **Response to Reviewer kDeW Part (2/3)**
>
> > What are the limiting factors of the chosen regularization in the latent space?
>
> Our proposed regularization imposes constraints on the decoder network, $q$, using the task-relevance measure, $f$. It is limited by the availability of an appropriate function $f$ and implementation of constraint enforcement. More specifically:
>
> An energy function is required to formulate our diversity objective. To use our diversity objective with imitation learning (IL) approaches that do not explicitly model expert occupancy [1, 2], one must construct an energy function in some manner. We believe that $Q$ functions learned during policy optimization may act as suitable candidates and leave testing this hypothesis for future work.
> Constraints are approximately enforced using Lagrange multipliers. While the approximate enforcement allows exploration of the state space, it may allow spurious behaviors that drastically vary from demonstrations. Parametric approaches akin to spectral normalization for Lipschitz continuity [3] are desirable.
>
> We will update the extended limitations section in Appendix E.
>
> > How much is the representation limited by demonstrations?
>
> The latent representation is inherently guided by the demonstrations. Heterogeneity observed in demonstrations indicates the dimensions along which learned behaviors can vary. If certain dimensions of variations are absent in demonstrations, then our approach may not capture those variations. However, we deem absent dimensions to be irrelevant to the task and out-of-distribution. We see our reliance on demonstrations as a strength, as they help choose relevant dimensions and generalize behaviors over them. We highlight that learned behaviors are novel as variations along latent dimensions can result in dramatically different behaviors (e.g., varying speed in HalfCheetah results in different gait patterns). Unsupervised RL [4] learns diverse behaviors but learned behaviors may not be useful towards task completion without finetuning or downstream use as low level skills. Diverse RL, i.e., unsupervised RL augmented with rewards [5], learns diverse task-accomplishing policies. However, in both cases, behaviors exhibit variations that may lack semantic meaning or not align with user expectations. Our approach addresses the challenging setting of learning diverse behaviors while inferring both behavior factors and task specifications directly from demonstrations.
>
> > Does this approach suffer from problems similar to behavior cloning?
>
> No, our approach does not suffer from the same issues as behavior cloning. While behavior cloning is prone to covariate shift, especially in low data settings [7], our approach leverages online inverse reinforcement learning, which explicitly addresses covariate shift through environment interaction [8]. Covariate shift may also be tackled with online expert interaction [9], but having human experts online to provide corrections may not be feasible in many settings.
>
> > Is exploration limited and does this have an effect on final performance?
>
> No, exploration is not limited by our framework and does not affect final performance. Exploration of the state-action space is handled by the reinforcement learning (RL) algorithm, which operates independently of our framework and can be substituted with any method to synthesize a policy.
>
> Exploration can be examined from two different perspectives: (1) Coverage of the state action space by various behaviors, i.e., $pi(.|., z)$ for various $z$, after learning, (2) Exploration of the state-action space during learning.
>
> For (1), we refer to our response above for the question, “How much is the representation limited by demonstrations?”. In short, yes, coverage is limited and it improves performance. Coverage of the state-action space is limited to those regions associated with dimensions which are identified to be varying among demonstrations. Limited coverage allows for more behaviors to be relevant to identified factors which improves task and factor recovery performance.
>
> For (2), our approach modifies the diversity and imitation rewards that are provided to an RL routine (in our case, PPO) that synthesizes a policy in parallel with decoder and discriminator learning. Exploration of the state-action space can be limited if the rewards are sparse and uninformative that the RL routine underperforms. We take measures to obtain gradually varying and expressive rewards, as detailed in Appendix C1 and common in prior work [10]. We employ these measures across all baselines to nullify their effect in the empirical evaluation.

---

> > ### Comment · Reviewer_kDeW · 2024-11-25
> >
> > Thank you for addressing my comments and providing the clarifications. I also appreciate the details provided to address the other reviewers' concerns.

---

> > > ### Author Response · Authors · 2024-12-02
> > >
> > > We thank you for your valuable time and feedback. As the end of the rebuttal period is nearing, we remind and invite you to discuss any pending questions, concerns or feedback to further improve our manuscript.

---

> ### Author Response · Authors · 2024-11-21
> **Response to Reviewer kDeW Part (3/3)**
>
> [1] Garg, D., Chakraborty, S., Cundy, C., Song, J., & Ermon, S. (2021). Iq-learn: Inverse soft-q learning for imitation. Advances in NeurIPS, 34, 4028-4039.
>
> [2] Haldar, S., Mathur, V., Yarats, D., & Pinto, L. (2023, March). Watch and match: Supercharging imitation with regularized optimal transport. In Conference on Robot Learning (pp. 32-43). PMLR.
>
> [3] Miyato, T., Kataoka, T., Koyama, M., & Yoshida, Y. (2018, February). Spectral Normalization for Generative Adversarial Networks. In the ICLR.
>
> [4] Laskin, M., Yarats, D., Liu, H., Lee, K., Zhan, A., Lu, K., ... & Abbeel, P. URLB: Unsupervised Reinforcement Learning Benchmark. In Thirty-fifth Conference on Neural Information Processing Systems Datasets and Benchmarks Track (Round 2).
>
> [5] Zahavy, T., Schroecker, Y., Behbahani, F., Baumli, K., Flennerhag, S., Hou, S., & Singh, S. Discovering Policies with DOMiNO: Diversity Optimization Maintaining Near Optimality. In The Eleventh ICLR.
>
> [6] Osa, T., Tangkaratt, V., & Sugiyama, M. (2022). Discovering diverse solutions in deep reinforcement learning by maximizing state–action-based mutual information. Neural Networks, 152, 90-104.
>
> [7] Osa, T., Pajarinen, J., Neumann, G., Bagnell, J. A., Abbeel, P., & Peters, J. (2018). An algorithmic perspective on imitation learning. Foundations and Trends® in Robotics, 7(1-2), 1-179.
>
> [8] Ho, J., & Ermon, S. (2016). Generative adversarial imitation learning. Advances in NeurIPS, 29.
>
> [9] Ross, S., Gordon, G., & Bagnell, D. (2011, June). A reduction of imitation learning and structured prediction to no-regret online learning. In Proceedings of the fourteenth international conference on artificial intelligence and statistics (pp. 627-635). JMLR Workshop and Conference Proceedings.
>
> [10] Orsini, M., Raichuk, A., Hussenot, L., Vincent, D., Dadashi, R., Girgin, S., ... & Andrychowicz, M. (2021). What matters for adversarial imitation learning?. Advances in NeurIPS, 34, 14656-14668.
>
> [11] Li, Y., Song, J., & Ermon, S. (2017). Infogail: Interpretable imitation learning from visual demonstrations. Advances in NeurIPS, 30.
>
> [12] Park, S., Choi, J., Kim, J., Lee, H., & Kim, G. (2022). Lipschitz-constrained unsupervised skill discovery. In the ICLR.
>
> [13] Orsini, M., Raichuk, A., Hussenot, L., Vincent, D., Dadashi, R., Girgin, S., ... & Andrychowicz, M. (2021). What matters for adversarial imitation learning?. Advances in NeurIPS, 34, 14656-14668.
>
> [14] Vaswani, A. (2017). Attention is all you need. Advances in NeurIPS.
>
> [15] Mathieu, E., Rainforth, T., Siddharth, N., & Teh, Y. W. (2019, May). Disentangling disentanglement in variational autoencoders. In the ICML (pp. 4402-4412). PMLR.
>
> [16] Tessler, C., Kasten, Y., Guo, Y., Mannor, S., Chechik, G., & Peng, X. B. (2023, July). Calm: Conditional adversarial latent models for directable virtual characters. In ACM SIGGRAPH 2023 Conference Proceedings (pp. 1-9).
>
> [17] Batra, S., Tjanaka, B., Nikolaidis, S., & Sukhatme, G. (2024, July). Quality Diversity for Robot Learning: Limitations and Future Directions. In Proceedings of the Genetic and Evolutionary Computation Conference Companion (pp. 587-590).

---

> ### Author Response · Authors · 2024-11-28
> **Manuscript Updated**
>
> We have revised the manuscript based on all the reviewers’ feedback. Revisions are highlighted in red. We list the updates made in response to your concerns below:
> - Improved scatter plot readability in Figure 5 by adding perpendiculars to axes and annotations that indicate directions corresponding to better performance.
> - Expanded limitations in Sec 7 (Lines 526-534) to discuss challenges with high-dimensional or non-Markovian latent factors, and other aspects related to scalability.
> - Added further limitations discussing extensions to non-adversarial IL approaches and concerns with approximate constraint enforcement in Appendix G (Lines 1306-1312).

---

### Meta-Review · Area_Chair_6TN2 · 2024-12-21

**Metareview:**

This paper presents a novel imitation learning approach that employs a diversity formulation to generate behaviors capable of generalizing across demonstrators' latent preferences. Empirical validation on three continuous control benchmarks demonstrates that the proposed method significantly outperforms baseline approaches in discovering novel behaviors. Furthermore, the authors showcase the method's ability to generalize striking behaviors for table tennis in a virtual testbed, leveraging human demonstrations collected from real-world scenarios.

The reviewers unanimously acknowledge the importance of the addressed problem, the novelty of the proposed method, and the strength of the evaluation using real-world data. Additionally, the paper is well-written, with clear and concise presentation. Therefore I am happy to recommend acceptance of this paper.

**Additional Comments On Reviewer Discussion:**

The reviewers had several questions regarding missing literature, discussion of limitation and the details of the algorithm design. The authors addressed all major concerns the reviewers had during the discussion period and updated the manuscript.

---

### Decision · Program_Chairs · 2025-01-22

Accept (Poster)